# Particle Number Concentration and SEM-EDX Analyses of an Auxiliary Heating Device in Operation with Different Fossil and Renewable Fuel

Péter Nagy [1], Ádám István Szabó [1], Ibolya Zsoldos [2] and György Szabados [3,*]

1 Department of Propulsion Technology, Széchenyi István University, Egyetem tér 1, H-9026 Győr, Hungary; nagy.peter@ga.sze.hu (P.N.); szabo.adam@ga.sze.hu (Á.I.S.)
2 Department of Materials Science and Technology, Széchenyi István University, Egyetem tér 1, H-9026 Győr, Hungary; zsoldos@sze.hu
3 Department of Mechatronics, Optics and Mechanical Engineering Informatics, Faculty of Mechanical Engineering, Budapest University of Technology and Economics, Műegyetem rkp. 3, H-1111 Budapest, Hungary
* Correspondence: szabados.gyorgy@mogi.bme.hu

**Abstract:** Pollution from road vehicles enters the air environment from many sources. One such source could be if the vehicle is equipped with an auxiliary heater. They can be classified according to whether they work with diesel or gasoline and whether they heat water or air. The subject of our research series is an additional heating system that heats the air, the original fuel is gasoline. This device has been built up in a modern engine test bench, where the environmental parameters can be controlled. The length of the test cycle was chosen to be 30 min. The tested fuels were E10, E30, E100 and B7. A 30-min operating period has been chosen in the NORMAL operating mode of the device as a test cycle. The focus of the tests was particle number concentration and soot composition. The results of the particle number concentration showed that renewable fuel content significantly reduces the number concentration of the emitted particles ($9.56 \times 10^8$ #/cycle for E10 vs. $1.65 \times 10^8$ #/cycle for E100), while B7 causes a significantly higher number of emissions than E10 ($3.92 \times 10^{10}$ #/cycle for B7). Based on the elemental analysis, most deposits are elemental carbon, but non-organic compounds are also present. Carbon (92.18 m/m% for E10), oxygen (6.34 m/m% for E10), fluorine (0.64 m/m% for E10), and zinc (0.56 m/m% for E10) have been found in the largest quantity of deposits taken form the combustion chamber.

**Keywords:** fuel-operated auxiliary heater; fossil fuels; renewable fuels; particle number concentration; EDX analysis




## 1. Introduction

In transport, the use of liquid fuels at the global level until 2030 and 2050, considering the forecasts made with different methodologies, may increase or decrease. Regardless, renewable liquid fuels are increasingly important [1]. However, there are also ideas regarding zero-emission transport [1,2]. Renewable fuels are mixed with traditional fossil fuels in many parts of the world, but e.g., also in the European Union [3]. Mixing renewable components into the original fuel of diesel engines into diesel has long been and is an area of extensive research. The same applies to the Otto engine and its initial fuel, motor gasoline, whose main renewable component is bioethanol [4–6]. Standards have been developed for fossil and renewable fuels in the European Union. Based on these, fuels have different physical and chemical properties relevant to combustion [7–12]. These properties determine the type of engine in which they can be used and affect engine operation and emissions. During the use of motor vehicles, there are many sources from which air pollution is realized. Air pollutants resulting from fuel use are limited to a limited value for both road and non-road vehicles according to UNECE (United Nations—European Economic

Commission) regulations (we do not deal with rail and water transport in this study). To comply with the limit values, engines are equipped with exhaust gas aftertreatment systems [13–15]. Other sources of pollution in a vehicle can be brake pad wear, tire wear, or the release of air conditioning into the environment. In the case of these, the release of pollution into the environment is not limited or treated [16–19]. There can be another source, the passenger compartment or engine coolant heaters, which, if they operate with liquid fuel, also release pollutants into the air untreated. There are two types of auxiliary heating: (i) one that heats the air and blows it into the passenger compartment and (ii) another that is integrated into the engine coolant circuit and heats the coolant [20–23]. The examination of harmful substances from heating devices did not start long ago, and more literature is needed on this topic. Among the tested pollutants are airborne and particulate contaminants. So far, among the relevant particles, the particle number has been investigated in most cases during a few experiments in such parking heaters [24–28]. There is extensive literature on particulate matter from internal combustion engines. Experiments have been conducted on diesel engines or vehicles [29–34] and Otto engines [35–44].

## 1.1. Particle Number and Special Fuel Tests on Auxiliary Heating Devices

In the study [25], the authors conducted experiments to study the emissions from the auxiliary heater on passenger vehicles in winter conditions, simulating real use. The selected cars were equipped with the vehicle's original heating equipment, including gasoline and diesel heating equipment. The transient phases, i.e., the start and stop phases, showed outstanding particle number peaks, while the concentration values were stable during the steady state phase. The particle number concentrations reached $590 \times 10^{12}$ #/kg$_\text{fuel}$ for gasoline heaters and $560 \times 10^{12}$ #/kg$_\text{fuel}$ for diesel heaters. A comparison of the number of particles emitted between the additional heating and the exhaust pipe emissions of the engine showed—for particles larger than 23 nm—that a typical heating cycle equal to the number of particles required to cover hundreds or even thousands of kilometres, the vehicle depending on the level of emissions. According to the authors, the high emission factors of heaters raise the question of whether it is justified to include the emissions of heaters in evaluating the total emissions reduction. The authors propose that the emissions of the auxiliary heating and the engine should be treated equally in the emission regulations. According to them, emission after-treatment devices would be a possible solution to the high emissions of fuel-powered AHs (Auxiliary Heaters).

According to the results presented in the literature [26], the additional heaters' particle number emissions are approximately three orders of magnitude higher than that coming from the exhaust pipe of idling gasoline vehicles. Similar to the previous study, for both gasoline and diesel AHs, the highest concentrations were shown immediately at the beginning and end of combustion. In contrast, the instantaneous concentrations were close to each other during steady-state operation, the duration of which depends on the user's preferences. Especially for diesel vehicles with an emissions rating after Euro 5b, all equipped with a particulate filter, the particle number emission levels from the engine tailpipe are very low compared to the emissions from the auxiliary heater. According to the authors, the results raise the question of whether the emission reduction is justified when using heaters.

According to [28], environmental protection issues are raised by the increase in plastic waste. Therefore, using alternative liquid as a fuel can be a research direction for sustainability. This work used the improved plastic pyrolysis oil to test an auxiliary heater with a heating power of 5 kW. The plastic pyrolysis mid-oil (diesel-like fraction, C11-C22) was similar to diesel regarding calorific value and viscosity. Mixtures were prepared from diesel gas oil and pyrolysis oil for the tests. The blends showed similar combustion and emission performance compared to 100% diesel combustion. CO (Carbon Monoxide), $CO_2$ (Carbon Dioxide) and $NO_x$ emissions were measured during the experiments; there was no particle number test. We have included this literature to show how wide the range of alternative

fuels that can be used for such heaters can be. According to the study, plastic pyrolysis fuel is a promising alternative to diesel in air heating applications.

Source [27] analyses and evaluates heaters that can be retrofitted to electric cars. Among the options available are fuel-operated heaters (FOH), one type of heater that heats the air. It compares the heating capabilities of each solution, focusing on cabin temperature and electrical consumption, and analyses the effect of each heating device on the electric car's range. Based on the results, he concludes that electric heaters significantly reduce the range of an electric car; thus, FOHs offer an attractive solution in cold regions. In addition, retrofitting presents no particular problems. During the well-to-cabin evaluation, it turned out that in the case of a vehicle with purely electric drive, $CO_2$ emissions appear if FOH is installed. In contrast, a vehicle with a purely electric drive has, in principle, zero emissions.

### 1.2. Particle Number Tests on Diesel-Engines

The development of this requirement in the type test was preceded by numerous experiments, but the experiments have been ongoing ever since. For example, with modern instruments, they can now go down to the size of 1.5 nm and count [29]. During a test cycle, the start-up and warm-up phases, which are a transient operation for most parameters, are very important. In the warm-up phase, the emission is significant compared to steady-state operation. If the steady-state phase is not too long, the emission realized during the warm-up phase represents a large proportion of the emission of the entire cycle. When a renewable component was mixed into diesel, an increased number of particles was observed during the warm-up phases [30].

In the research [31], the emissions of vehicles with different engines (Diesel and Otto) and fuel (gasoline and diesel) were examined in a predetermined driving cycle. The difference between the tests was at the start of the driving cycle, (i) with a cold engine, (ii) with a preheated cold engine, or (iii) with a warm engine. Changes were measured for PM (Particle Mass) and $NO_x$ (Oxides of Nitrogen) between the different starting forms but not for PN (Particle Number).

In a series of experiments published in the literature [32], the particle-relevant parameters, such as particle diameter, burned fraction and particle number emissions, were investigated on a diesel engine with the addition of renewable fuels, with the engine in a steady state. The renewable fuels were biodiesel and alcohol, with two and three mixtures of diesel/biodiesel and diesel/biodiesel/alcohol tested. From the particle number results, it can be concluded that the higher the renewable share in the tested mixture, the lower the number of emitted particles, regardless of whether the renewable substance is biodiesel or alcohol. The renewable materials achieve a significant reduction in the particle number in such a way that the cylinder pressure drop and heat release do not change significantly.

Research from [34] focuses on cold start and renewable mixing, and the two most important investigated parameters are particle number and particle size distribution. It examines 10, 15 and 20% mixtures mixed with renewables on a volumetric basis. The renewable material is coconut-oil-based biofuel. The tested engine was a six-cylinder supercharged diesel engine with a common rail fuel supply system. Four different launch phases were experimented with. The results are consistent, and in each start-up phase, the emitted particle number increases up to a certain renewable mixing ratio (0–15 V/V%) and decreases to some extent compared to the 15% result for the last mixture (20 V/V%). The amount of reduction depends on the applied start-up phase. The decrease is smaller for the two cold starts, and the drop is more significant for the two warm starts.

### 1.3. Particle Number Tests on Otto-Engines

Soot particles in the flame and exhaust gas were examined in a cylindrical, optical, spark-ignition, direct injection engine. A unique thermophoretic sampling technique was used for sampling from the flame. The number of soot particles and other parameters were analysed using images recorded on a TEM grid. The tested fuels were motor gasoline and various ethanol mixtures. A significant decrease in the number of particles was observed

with the increase in ethanol incorporation. That was true both in the flame and in the exhaust. Ethanol had a more significant effect on particles in the flame than on particles in the exhaust gas. Thus, the reduction in particle number is much more significant in the flame than for the particles in the exhaust gas [35].

One study [36] investigated the effect of mixing ethanol on a spark-induced compression ignition engine. The tested fuels were RON93, RON80+E15 and E100. RON93 and RON85+E15 were used for the SICI (Spark Induced Compression Ignition) process, and E100 was used for the SI (Spark Ignition) process. In addition to the particle number, several operating parameters were examined. PN emission showed a positive correlation with knock intensity. A lean mixture plant is advantageous in terms of particle number. Furthermore, adding ethanol significantly reduces the PN emission, which is explained by the higher oxygen content and the presence of more reactive OH and $HO_2$ radicals.

Ethanol–gasoline blends of 10, 25 and 85% V/V% were tested on a direct injection turbocharged engine during the transient cycle (WLTC). The particle number was investigated but for particles smaller than 23 nm in size. Mixing ethanol into the tested fuel had a particle-reducing effect. An opposite trend was obtained during cold start, when due to the lower heating value of ethanol, more fuel had to be injected, which worsened the mixture formation and caused wall wetting, which is favourable for particle formation. Particles below 23 nm in size represent a significant fraction of particle emissions. The size of the fraction of sub-23 nm particles increases when the ethanol content in the mixture increases [37].

The solid particles in the flame were tested on a direct injection gasoline engine by increasing the ethanol content (E10, E30 and E50) in the tested fuel. A fibre optic sensor was used for this. The engine operated with two types of mixture formation: (i) lean-homogeneous mixture formation and (ii) lean-stratified mixture formation. The lower power available with ethanol required more ethanol to maintain the same engine power, which increased particle number emissions. Fuel properties significantly influence particulate emissions more than engine operating parameters [41].

Particle number emissions of vehicle engines were investigated on three vehicles with different mixture formation systems. These were PFI (Port Fuel Injection), GDI (Gasoline Direct Injection) and MI (Mixed Injection), which is a combination of the first two. The WLTC cycle was chosen as the vehicle test cycle. The emission classification of the vehicles was the same. The MI vehicle showed much lower particle number emissions in the low-medium and high-speed phases than the other two. Only commercially available motor gasoline was used during the tests, and no renewable fuel was used [42].

*1.4. Some Emission and Atmospheric Relevant EDX Analysis*

The EDX (Energy-Dispersive X-ray) method is a spectroscopic method suitable for determining elements with a higher atomic number than boron. This method is unsuitable for detecting hydrogen and hydrocarbons because it is ideal for testing organic ingredients and inorganic components [45]. There are examples of the analysis of particles from diesel engines and gasoline engines. The analysis of the soot particles produced during the diesel engine's combustion showed that the particles mainly consist of carbon and oxygen, but hydrogen can also be found. Sulphur can also be detected [46].

Research [47] compared the soot emissions of a heater and a direct injection gasoline (GDI) engine with many tests, including EDX tests. GDI soot EDX results were taken from literature sources. It was determined that the burner and GDI soot gave almost identical results and that the soot was carbon in nature.

Tests have also been carried out to analyse, for example, lubricating oil or atmospheric aerosol composition [48–50].

*1.5. Some Other Aspects of Using a Fuel-Operated Auxiliary Heater*

Additional fuel-powered heating devices are widely used in passenger cars, freight transporters, heavy-duty vehicles, and sailboats. They act as a stove during their operation,

generating heat by burning fuel. The exhaust gas from the combustion process of the devices is not treated afterwards, so it goes directly into the environment. The devices can be grouped based on the type of fuel burned and the heated medium. Since the heater usually does not have a separate fuel tank, it is connected to the fuel tank of the respective vehicles. Therefore, depending on the engine type, they manufacture heating devices that run on gasoline or diesel. The product range of some manufacturers already includes equipment that works with renewable fuel. Based on the heated medium, they can be divided into two main groups. Devices in the first group heat air for heating the cabin or loading space. The devices belonging to the second group are connected to the cooling system of the vehicle's engine, thus heating the engine's coolant. That is necessary so that the heating system can send warm air into the vehicle's passenger compartment in a short time. However, thanks to this, the engine's warm-up phase is reduced, thus reducing the number of pollutants emitted during the warm-up phase and improving the cold start properties for the critical parameters related to cold start. Depending on the use area, these devices are used seasonally or year-round [20,21,51].

The counting of particles emitted by road vehicles has been included in the type test requirements with the EURO 5b regulation in the case of passenger cars and the EURO VI regulation in the case of engines of heavy-duty vehicles. Based on this, the number of solid particles with a diameter greater than 23 nm (CMD, Count Median Diameter) coming out of the engine had to be determined and compared with the limit value [52,53]. The UN-ECE regulation also applies to the harmful emissions of the heating equipment, but the particle number is not regulated. CO, HC, $NO_x$ and Bacharach number are given as regulated components [54]. Study [55] deals with analysing solid particle emissions from heating equipment. It highlights the importance—considering the European directives—that the number of renewable components to be mixed into the motor gasoline used in Europe will change, affecting the emissions of such devices.

In their previous article [56] on harmful emissions from heating devices, the authors of current manuscript examined the emission of gaseous components of such devices. Based on the measurement results, a straightforward model was used to calculate the total emissions of a vehicle fleet (the size of Hungary) equipped with such a device. That shows that about certain components, e.g., $CO_2$, the emission is not significant. However, it concerns other components, e.g., THC (Total Hydrogen Carbon), which is directly harmful to the environment.

### 1.6. The Aim of This Research

There needs to be more literature on the measurement of particle number emissions from passenger compartment heaters, as presented in detail above. No experimental results for such devices can be found for ash testing. Our current work aims to measure and analyse the particle number emissions of a gasoline-powered heater and the dry matter content taken from the mesh surface of the atomizer by operating the device with gasoline and gasoline/bioethanol mixtures of various proportions, as well as with diesel. The tests with diesel oil are carried out for the following reasons. We were curious whether the gasoline-powered device would run stably with diesel. Some sources state that the construction of gasoline-powered and diesel-powered devices is similar [57,58]; therefore, in principle, the heating device can also work with diesel. The second is what kind of results can be obtained about the above two parameters (particle number, elemental analysis of dry matter) with diesel oil. In our case, it is an air-heating device that works with liquid fuel. We want to emphasize that we only examine short-term results and immediate effects when operating with bioethanol and diesel, which are not intended for the device. We do not examine the long-term effects of using bioethanol or diesel on the device. Our goal with the analysis is twofold: on the one hand, we try to describe the processes and their background in an acceptable way from a scientific point of view, and on the other hand, we also analyse the simple indicators formed from the processes.

## 2. Materials and Methods

In the present research, for analysing the particle number emission of the heater, we used the commercially available E10 fuel as the base fuel. Then, we measured by increasing the volume percentage composition of this bioethanol. The base measurements were performed with (i) E10 fuel containing 10 V/V% bioethanol [8,12]. We obtained this from the fuel station of a Hungarian oil refinery in Hungary. The bioethanol used as a mixing component is a renewable fuel of vegetable origin, produced in a factory in Hungary specifically for transport purposes, and we obtained it from there. The purchased fuel meets the requirements of the relevant EU standard [9]. In addition to the base fuel, we examined two fuels. The first was an increase in the volume percentage content of bioethanol in the fuel, a mixture containing (ii) a 30 V/V% renewable component and (iii) pure bioethanol, which did not have a fossil component. The original fuel of the stationary heating device is motor gasoline. After comparing the device types, the authors concluded that gasoline and diesel-powered devices do not differ from each other structurally, or only to a small extent, so they were curious as to whether the device optimised for operation with gasoline can also operate with diesel, and if so, to what extent the combustion process and particle number emissions change. Therefore, the fourth tested fuel (iv) was diesel gas oil, obtained with motor gasoline from a fuel filling station. The purchased diesel also met the relevant European Union standard [7,11]. Table 1 summarises the investigated fuels.

**Table 1.** Investigated fuels with their standards.

| Investigated Fuel | Relevant Standard |
| --- | --- |
| E10 | EN 228 [8] |
| E30 (it is a mixture of E10 and E100 on a volumetric basis) | EN 228 and EN 15376 [9] |
| E100 | EN 15376 |
| B7 | EN 590 [7] |

The measurements were carried out in a modern engine test bench. The temperature and humidity of the room can be controlled. We chose t = 15 °C and relative humidity: 30% because these were the smallest values that the air handling equipment could stably produce during all measurements. The air with the above parameters was the intake air of the heating device and the air surrounding the device and fuel system. The adjusted and controlled parameters are necessary for the repeatability of the tests. The stationary heating device can operate in three pre-programmed modes: the ECO-NORMAL-BOOST modes. We chose the normal mode, assuming that the device is used this way in most cases. In the case of all measurements, the device was operated for 1800 s from start-up, which period was chosen because, according to the manufacturer's statistics [20], users usually use the machine for that long during a start-up. After the measurements, the equipment was tempered before starting the measurement again to ensure that the starts were always under the same conditions. Before beginning the measures, a self-cleaning program was run on the measuring devices. We performed three measurements with each tested fuel and then averaged their results. We did not calculate the standard deviation from the three measurement results.

The device was wholly disassembled and cleaned before each fuel measurement series, and a new burning mesh was installed. After the cycle, when the machine had cooled down, we removed the burning mesh, took photos of them and took dry material samples from their surface, which samples were subjected to SEM-EDX (Scanning Electron Microscopy—Energy-Dispersive X-ray) analysis (elemental analysis). The sampling of dry matter was not carried out based on a standard; we did not use a standard for that. Each different soot sample was tested at five randomly chosen points in EDX. The measurement results were processed in Excel and calculated from their average and standard deviation.

From the results obtained, we made different indicators at the end of the study, based on which we made further evaluations. That is the first result of its kind, obtained with the particle count/kg fuel unit, with which we want to compare fuels. The second is the particle number/tested cycle, with which we wish to obtain approximate emission values. What value should be calculated if such a device is operated for half an hour? The third indicator is created to have the unit of particle number/km using [52], and then we plan to compare it with the prescribed limit value found in the emission type test of road passenger vehicles [52].

### 3. The Experimental Set-Up

The device was built on a frame in an engine test bench to examine the number of particles in the device's exhaust gas and to analyse the processes in the combustion chamber. Figure 1 shows three separate images. The left part shows a schematic layout. In the middle part is a photograph showing the test frame during preparation, with the main components. On the right side of the figure, a picture shows the heater testing frame placed in the engine test room. The make and type characteristics of the elements are listed in Table 2.

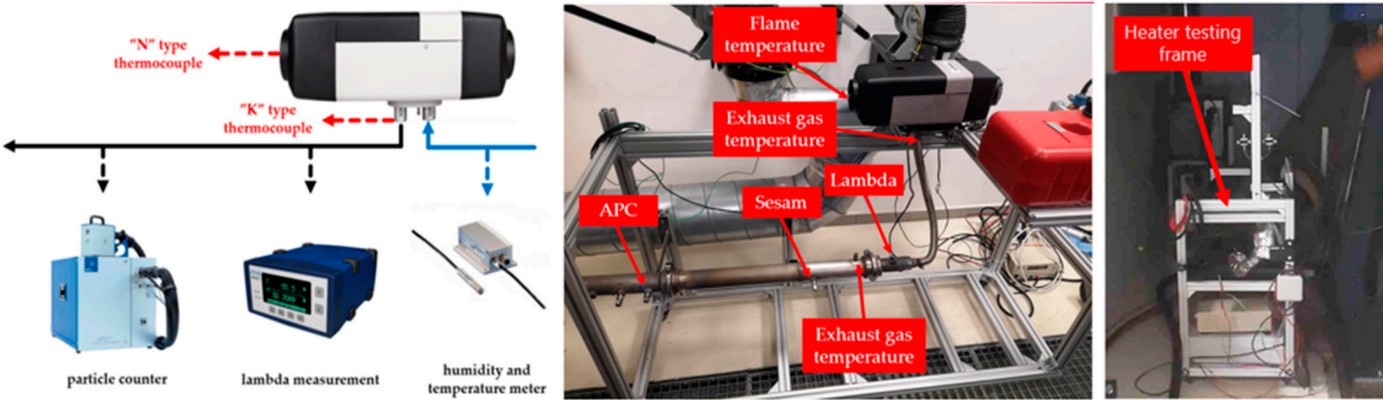

**Figure 1.** The experimental set-up [own recording, editing].

**Table 2.** Parts of measurement system for exhaust and chamber analysis [own editing].

| Path | Parameter | Instrument, Device | Make, Type |
|---|---|---|---|
| Air | Intake air humidity and temperature | Humidity and temperature sensors | Vaisala HMT310 |
| Combustion | Flame temperature | Thermo couple | N type sensor with QuantumX MX1609B |
| | Air excess ratio | Lambda sensor | Bosch LSU 4.9 wide band sensor with ETAS ES636.1 module |
| Exhaust | Exhaust temperature | Thermo couple | K type sensor with QuantumX MX1609KB |
| | Exhaust particle number | Particle counter | AVL Particle Counter: APC 489 |

The Particle Number Counter (APC) was the main component in this measurement system. That is an AVL-manufactured instrument of the type AVL APC 489. It operates according to the principle of Condensation Particle Counting (CPC). It counts the non-volatile particles with greater CMD than 23 nm [59]. Programs of AVL PUMA 2TM [60] and Webasto Thermo Test [61] have been used for data acquisition. The programs AVL CONCERTO 5™ [62] and Microsoft Excel program have been used to plot the acquired data. Calibrations of the instruments were performed according to the annual calibration plan for the laboratory and are valid. The heater and its measurement system built on it were placed in a state-of-the-art engine test bench, where the ambient air and the air sup-

plying the engine's combustion can be controlled regarding three parameters: temperature, pressure and relative humidity. The central part of the test bench's infrastructure is an AVL ConsysAir 2400 [63].

Elemental quantification tests were carried out with the help of a Hitachi S-3400N (Hitachi Ltd., Chiyoda, Tokyo, Japan) SEM. Each soot sample produced by burning fuel with different ethanol content has a unique elemental composition. The elemental quantities can be deduced using the additional energy-dispersive X-ray spectroscopy equipment (Bruker, Billerica, MA, USA) belonging to the SEM. Soot samples were stored at room temperature (22 ± 2 °C and 20 ± 5% RH) in well-closed glass vials for a required period before examination. Measurements were evaluated with the software ESPRIT 1.8, Bruker [64]. During the tests, the EDX spectrum of the soot was recorded, from which the elemental composition was determined by quantification calculation after the background deconvolution. To examine the EDX spectrum of the soot samples, the K$\alpha$ lines of the characteristic X-ray spectra of the elements were investigated. The ingredients present in the soot sample could be separated from each other in the spectrum by adjusting the width of the studied area. Figure 2 shows the two steps: sampling and analysing.

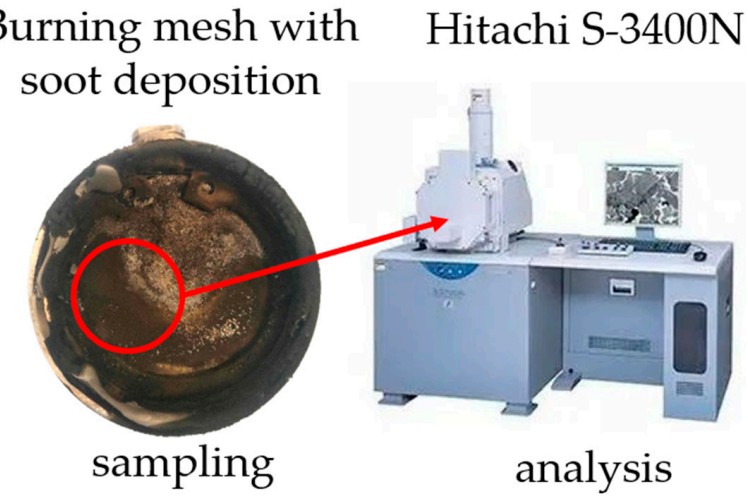

**Figure 2.** Place for sample of dry matter content and analysis [own editing].

## 4. Results and Discussion

Based on the literature, it can be said that the number of particles produced during combustion in internal combustion engines depends on many things, e.g., the fuel injection pressure, the atomization quality, the type of fuel used, the type of cycle, whether the engine is in a transient state or steady state, or the excess air factor. But it also depends on whether there are moving parts in the machine or not [24–44]. For this reason, we started our work by mapping the mixture formation system of stationary heating since mixture formation is a significant determinant of harmful emissions. There are few research results regarding the particle number testing of additional heating devices. There are two pieces of literature [25,26], but they do not investigate the effect of renewable fuel on the particle number. In Section 4, Results and Discussion, we can refer to these two results and results that can be found based on similar tests of internal combustion engines.

### 4.1. General Description of the Mixture Formation of Stationary Heaters

Figure 3 shows the structure of stationary heaters. The most critical components from the mixture formation point of view are numbered in the figure. Their tasks are described below.

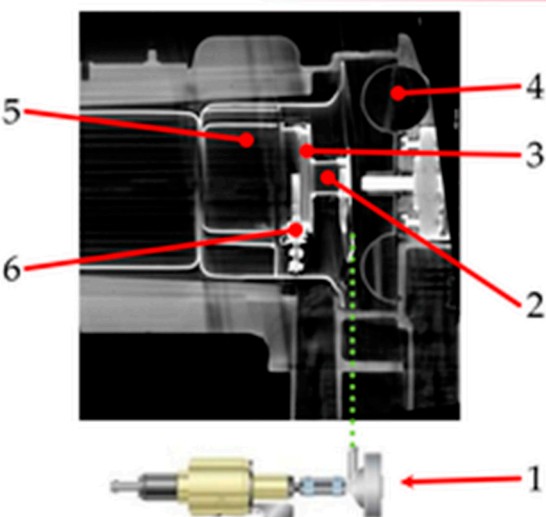

**Figure 3.** Construction of a fuel-operated air heater and relevant parts for mixture formation [own recording, editing].

1 Fuel pump: This frequency-controlled diaphragm pump delivers fuel drawn from the tank to the heater. During the start-up and burn-out phases of the device, the frequency is variable; in the steady state, the frequency is constant. The pump provides simple volume delivery. The fuel enters the chamber marked with the number 2 in a pulsating flow.

2 Fuel chamber: Inside the stationary heating device, fuel is placed in the chamber located on the "cold" side of the burner basket, the pulsating flow of which is dampened by a felt-like material placed in the chamber in front of the burning mesh marked with the number 3. The purpose of the damping insert is to provide as continuous a supply of fuel as possible for continuous combustion.

3 Burning mesh: This compressed metal mesh provides a boundary between the fuel chamber and the combustion chamber, thereby preventing backburning towards the outside. On the other hand, the fuel also performs atomization tasks. The fuel in the fuel chamber seeps through the mesh and disperses and evaporates in the upper part of the combustion chamber, where it feeds the combustion.

4 Fan: The fan motor that feeds the combustion has a dual role. On the one hand, it flows air with a higher mass flow around the externally ribbed surface of the device, which supplies the heating air. On the other hand, it delivers air with a lower mass flow into the combustion chamber with the help of a fan blade with axial and radial blades on a common axis. Since the device does not generate a suction effect like a motor, installing such an air carrier is necessary.

5 Combustion chamber: Air flows in through the perforated plate, delimiting the combustion chamber, which promotes fuel–air mixing in the combustion chamber by creating a turbulent flow and feeding the combustion.

6 Glow plug: This preheats the combustion chamber when the device starts the combustion process. When the machine is stopped, it starts working again, during which it burns the remaining fuel in the burning mesh. If an analogy can be used, it is a device with positive ignition, in which it is only necessary to light the flame once.

Based on the above analysis of mixture formation, the additional heating device, approx. It differs from an internal combustion engine in all the abovementioned parameters. The heater is an open-system, continuous combustion, heat-producing machine. Due to the simplicity of the mixture formation, we can expect relatively higher emissions than with an internal combustion engine.

### 4.2. Bioethanol's Effect on the Particle Number Concentration

The stationary heating device does not have a sensor to identify what fuel is being used. As a result, pre-written programs run throughout the operation, so in the case of all three tested fuels: gasoline, gasoline–ethanol and ethanol, e.g., the fuel delivery parameters are the same. The diagrams show an almost identical character in the case of the change in the number of particles as a function of time; only a quantitative difference is visible depending on the fuel used (Figure 4). We will not describe the detailed analyses of the individual phases now; we will deal with them in Sections 4.2.1–4.2.3. In short, the first big spike in emissions belongs to the start-up phase; there are hardly any emissions in the steady-state compared to this, and there is also a more significant peak in the burn-out phase.

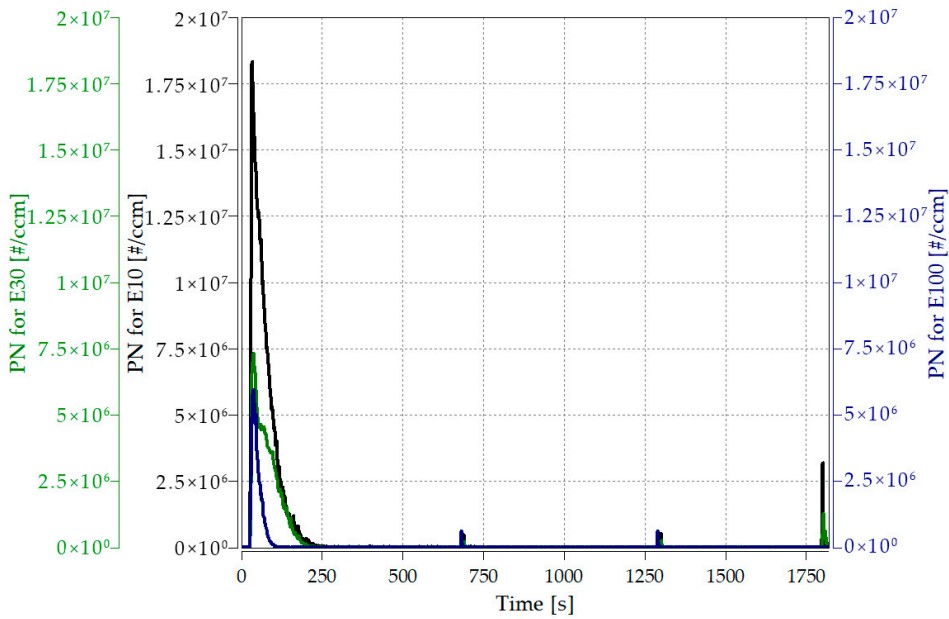

**Figure 4.** Particle number concentration of fuels (E10, E30 and E100) during the cycle.

During the tested cycle, the device generated emissions of all the particle numbers shown in Figure 5 with the different tested fuels.

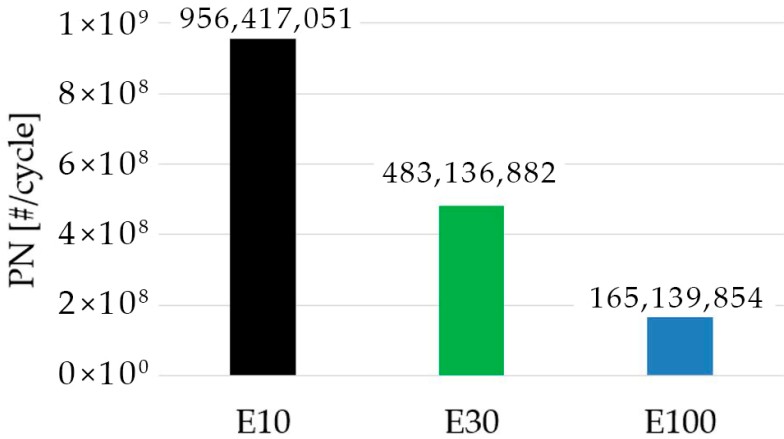

**Figure 5.** Numbers of total particle emission for the different fuels.

A significant reduction in particle emissions can be observed with an increase in the V/V% of the bioethanol content due to "cleaner" combustion. Regarding total emissions, the total number of particles measured during the burning of pure ethanol (E100) is only 17.3% compared to the measurement of the base fuel E10. Even at a mixing ratio of 30 V/V%,

which is not considered too high, the particle number is approximately halved over the entire cycle.

To make the differences between the values measured during the cycles visible as a function of time, in the following subsections, we present and analyse the results in detail in the three operating stages (during the entire cycle, start-up stable operation and the shutdown stage).

Regarding the concentration as a function of time, similar results were obtained in [25], which means that the three operating phases of such a device can also be distinguished using the particle number emission. The start-up and burn-out phases have a transient emission, and the steady-state phase has a uniform emission.

### 4.2.1. The Start-Up Phase

We call the warm-up or start-up phase the processes that take place from the start of the device until the combustion stabilizes. In this stage, the combustion chamber is heated using a glow plug, and fuel and air delivery begins. With the start of combustion, it is worth examining the lambda change first for the three fuels (upper part of Figure 6). In the case of all three fuels, the start control of the device was the same (lower part of Figure 6). After switching on, only the fan works at low speed for more than 25 s. From the 27th second, the fuel delivery starts, and the speed of the air fan also increases. The start-up enrichment (fuel pump operation at 6 Hz) only lasts for a short time, then decreases back to 2 Hz operation while the amount of air increases. By the end of the start-up enrichment, the lambda value is between 2.5 and 3.5, depending on the fuel. After about 1.5 min, the fuel delivery starts to increase, with it, the air delivery as well. It takes approximately 4.5 min when the fuel delivery will be at a constant value, but the air delivery will still increase. It can be seen that the device achieves a constant fuel supply frequency of around 280 s and the constant combustion feeding air supply, at which time the combustion process stabilizes. In the stable operating phase, the value of λ is between 1 and 2, depending on the fuel. The chemical symbol of ethanol is $C_2H_5OH$, so there is oxygen in the molecule. The stoichiometric air requirement of ethanol is lower than that of gasoline. Therefore, in theory, less air would be enough for combustion, but the air sucked in by the machine is always the same since the fan delivers the same amount. For this reason, the air condition factor that can be measured after combustion of the mixture containing the renewable component is higher than the one holding the fossil.

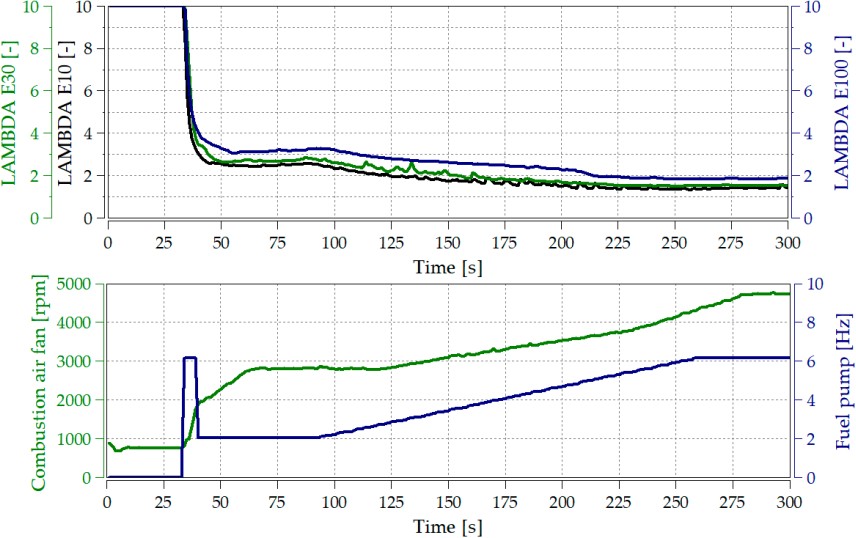

**Figure 6.** Variation of lambda values (**upper part**) with increasing ethanol content and fuel pump and air fan control signals (**lower part**) during the start-up phase.

The large number of particles that can be measured during the launch can be traced back to several things:

- When the fuel pump is started with a high frequency, fuel is introduced in a shock-like manner when the fuel droplets cannot evaporate sufficiently through the still-cold evaporator.
- At that time, the glow plug located at the burner basket was still working, near which the fuel droplets could not mix appropriately with oxygen, and near its hot surface, they burned by diffusion combustion.
- The glow stick only heats the device directly in front of the burner basket; in the initial combustion phase, the flame goes out near the walls of the cold combustion chamber.

During the start-up performed with each fuel, there is a significant difference in the particle number as a function of time. The bioethanol plant causes the smallest number of peak emissions among the three. That is a third of the E10 peak. Interestingly, in the case of the mixture with a small renewable mixing ratio (E30), the number peak is much closer to bioethanol's rise than diesel's peak value. The process can be seen in Figure 7. Figure 8 shows the evolution of total particle emissions that can be measured during the start-up in the first 280 s.

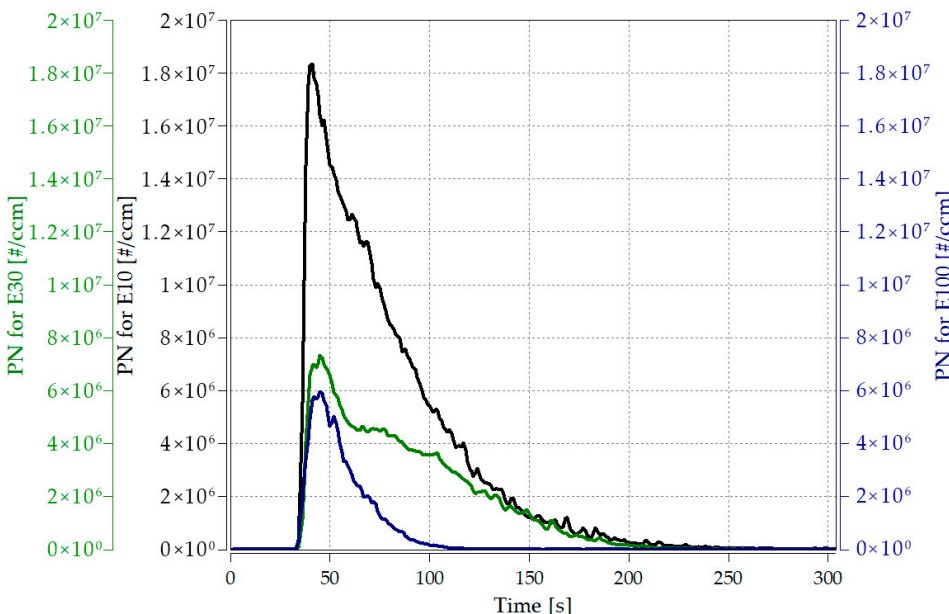

**Figure 7.** Particle number concentration changes during the start-up phase.

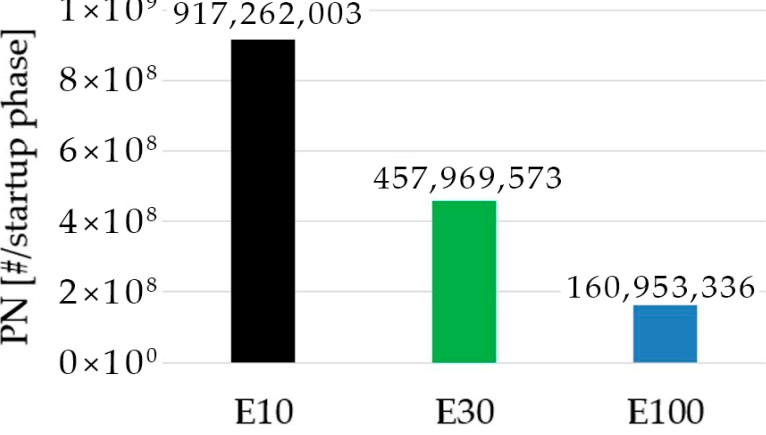

**Figure 8.** Numbers of total particle emission for the different fuels in the start-up phase.

During the cycle, the highest particle number emissions are in the start-up phase, and this is where the most significant variation is seen for different fuel mixtures. The sum of the number of emitted particles in the start-up phase (1800 s vs. 280 s) accounts for approximately 95% of the total emission for the entire test period. Using pure ethanol, only 17.55% of the total particulate emissions in the start-up phase are compared to the base fuel. In the case of the three tested fuels, the total particle emissions of the start-up phases correspond to approximately 90 h in the case of E10, 70 h in the case of E30, and 295 h in the case of E100.

### 4.2.2. The Steady-State Phase

In a steady state, after the stabilization of combustion, with almost constant fuel and air delivery, the average lambda values are as follows: E10—1.38, E30—1.5, E100—1.85. With the increase in V/V% of bioethanol, it can be observed in Figure 9 that the lambda, and thus the combustion, is more uniform in the case of E100 than in the case of the base fuel. According to the authors, the uniformity of the combustion can be primarily attributed to the composition of the fuel used. While motor gasoline consists of many chemical components, the composition of ethanol is chemically homogeneous. In this operating state, the stationary heating device's control unit reduces the pump's fuel delivery for 12 s every 610 s at a constant fan speed. As a result, excess oxygen is temporarily available during combustion, so the lambda value increases. At the same time, an outlier can be seen in the particle emission (Figure 10).

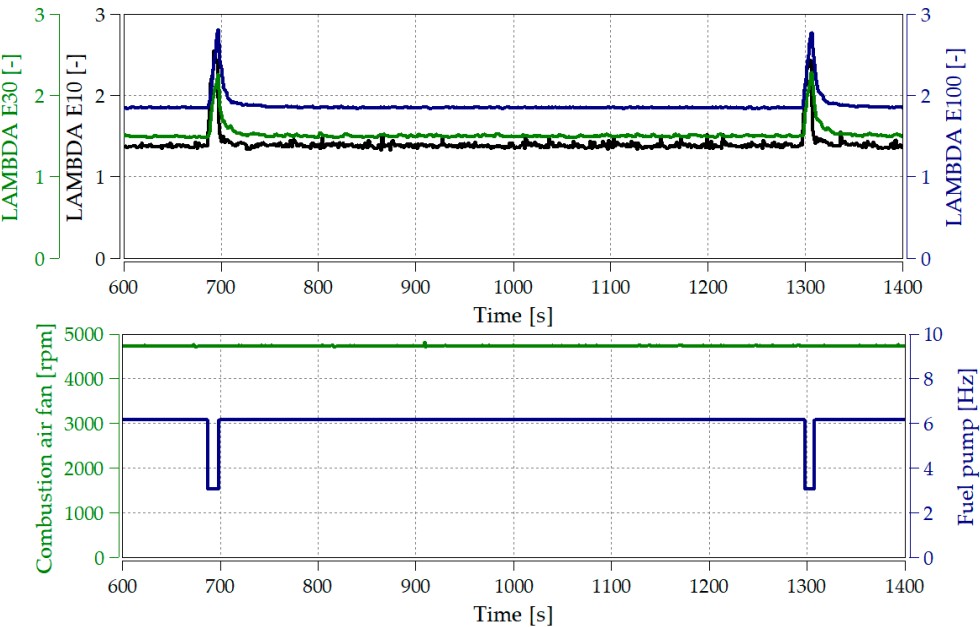

**Figure 9.** Variation of lambda values (**upper part**) with increasing ethanol content and fuel pump and air fan control signals (**lower part**) during the steady-state phase.

The momentary increase in particle emissions starts with more excess air being created by taking fuel, which causes a temporary change in the combustion process. That changes the flame pattern, and according to the temperature data, flame extinction begins because the flame temperature in the combustion chamber drops by nearly 140 °C during this time. The lambda value increases to 2.1 for E10, 2.3 for E30, and 2.8 for E100 during the fuel delivery return—the lambda value's upper limit in terms of gasoline ignition. A large number of particles are created when the fuel delivery is switched back on when the mixture is enriched. In a steady state, compared to E10 fuel, the E30 mixture results in almost 35% less particle number emissions and E100 95% less. More uniform combustion can also be observed in the particle emission, and with the increase in the ethanol content, a minor fluctuation can be seen on the diagram. Figure 11 shows the total values of the

particle number in the stable operating phase. The trend between fuels is the same as in the previous case; thus, E10 causes the most emissions and E100 the least. Overall, 67.5% of the emissions obtained with E30 and 5.8% received with E100 are the number of particles that can be measured by burning E10 fuel.

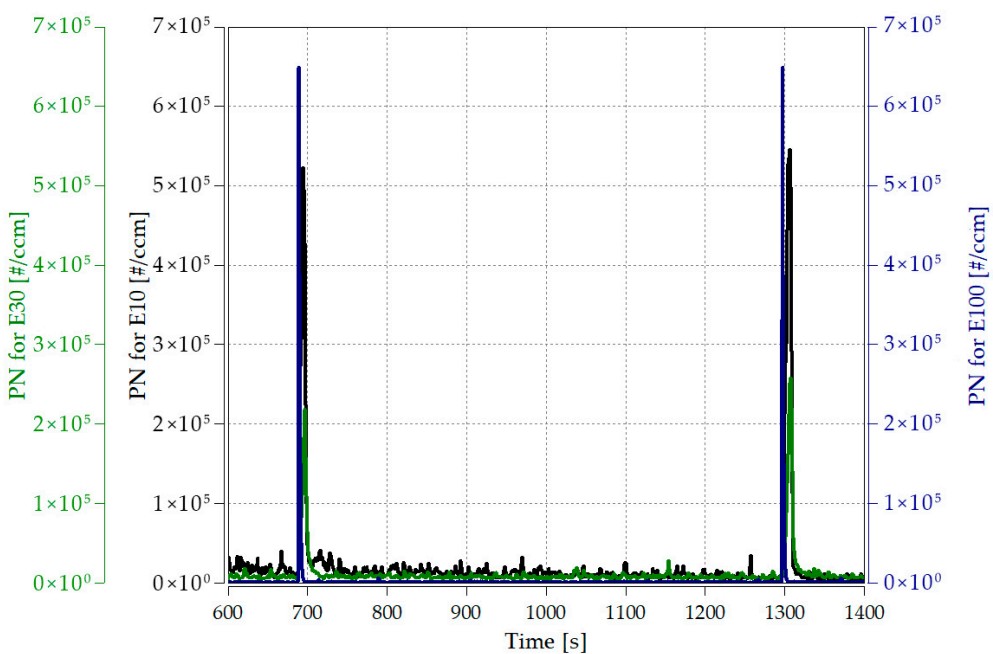

**Figure 10.** Particle number concentration changes during the steady-state phase.

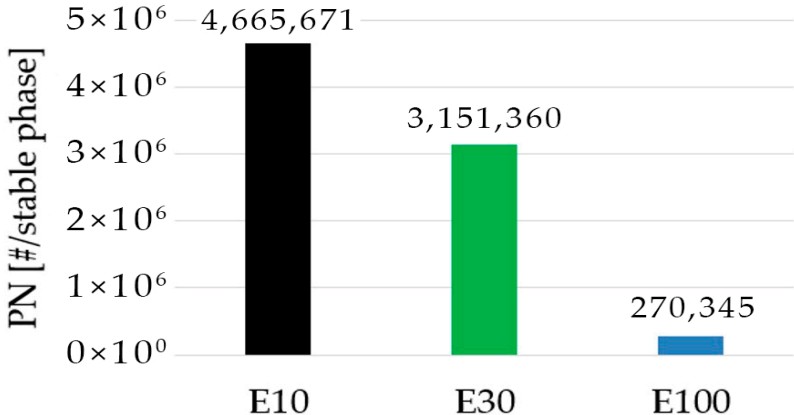

**Figure 11.** Numbers of total particle emission for the different fuels in the steady-state phase.

### 4.2.3. Burn-Out Phase

After stopping the device, the fuel delivery ends, the flame goes out, and the glow stick starts working again after a cycle stopped at 1800 s, approximately measurable particle number emissions for 10 s with the base fuel.

After stopping, the pump stops quickly, and the fan's speed decreases. The air factor values also increase. The lambda functions of E10 and E30 rise immediately after switching off; the value of E100 shows a delay of 5 s before it reaches value 3 set here. This process is shown in Figure 12.

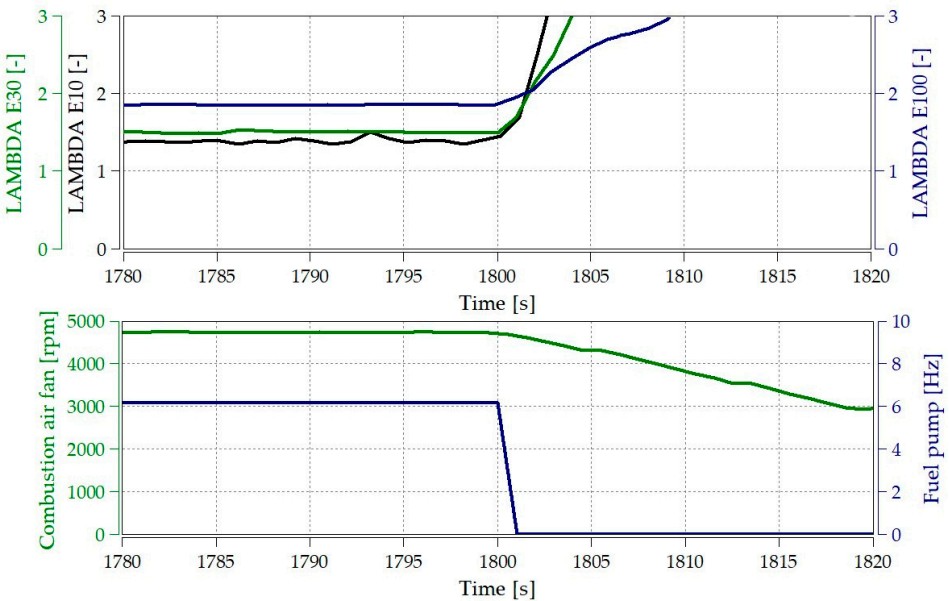

**Figure 12.** Variation of lambda values (**upper part**) with increasing ethanol content and fuel pump and air fan control signals (**lower part**) during the burn-out phase.

Then, the fuel remaining in the vaporizer is burned out, which is a transient process. Excess particle emissions are produced during non-corroded combustion (Figure 13). In the case of E100, it does not show a significant increase compared to the soot emission of the burn-out phase compared to the stable operating state; in the case of the E30 mixture, this value increases by 170 times compared to its average value measured in the steady operating condition, and by 275 times in the case of E100. Due to the scale, the number-time function of the E100 particle is not visible in the figure. With the two tested fuels containing more renewable material, "burn-out" cannot be achieved with a steeply rising and falling peak. For these, it takes longer to evaporate the remaining amount, which may be due to the higher heat of vaporization of bioethanol than gasoline [8,9].

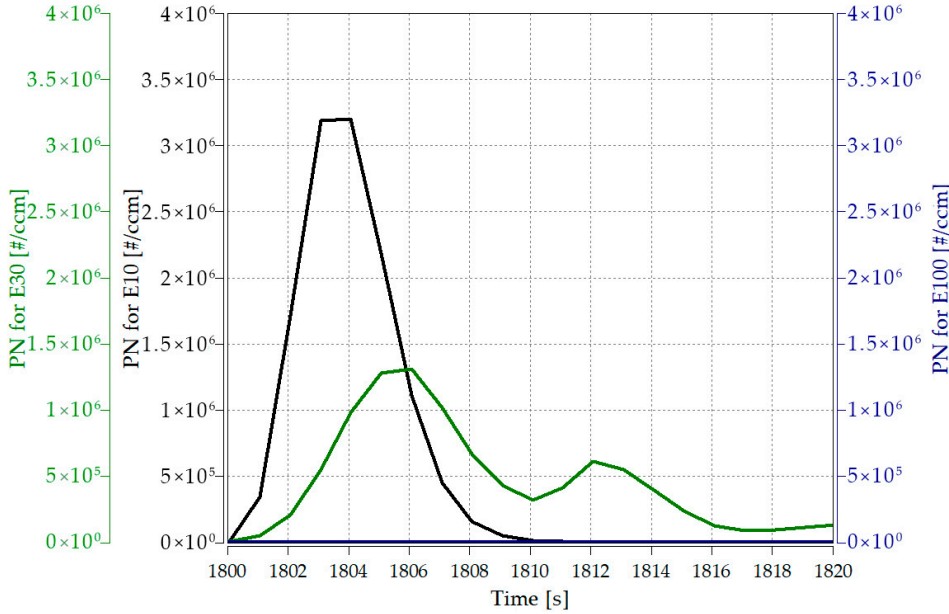

**Figure 13.** Particle number concentration changes during the burn-out phase.

The bar graphs in Figure 14 show the values of all the numbers in the burn-out phase. The specific numerical values are shown in the diagram. The trend in emissions caused by fuels is the same for the previous phases and the trend of the entire cycle. The emission corresponding to E30 is 85.3% of the value corresponding to E100, and the particle number realized by E10 is 0.4% of the value corresponding to the base fuel.

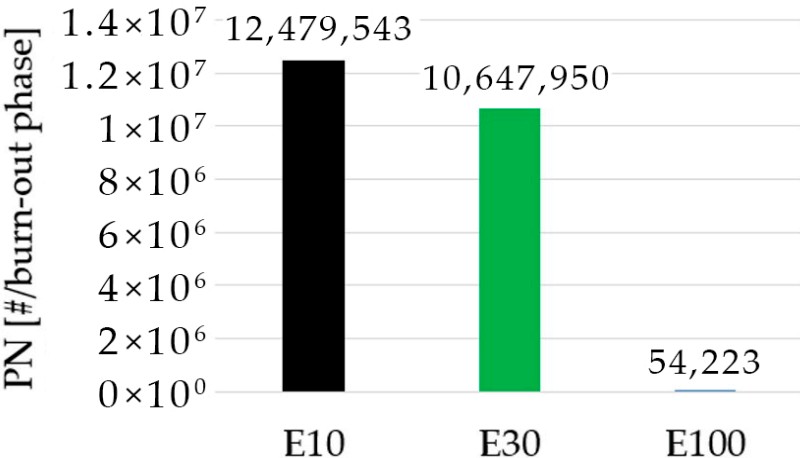

**Figure 14.** Numbers of total particle emission for the different fuels in the burn-out phase.

If we compare the magnitudes of each phase's numbers, the highest value for all tested fuels is in the start-up phase. The value obtained in the start-up phase in the case of E10 is two orders of magnitude higher than in the burn-out phase and three orders of magnitude higher than in the stable operating phase. The same type of change applies to the E30. For E100, the total number of particles in the start-up phase is four orders of magnitude higher than in the steady-state phase, and the value obtained in the latter is one order of magnitude higher than in the burn-out stage. The start-up phase is thus the most polluting of the three phases, with values corresponding to the currently used cycle time. This cycle time can be used for morning heating, but also for night heating; the usage time can be 10–20 times this.

Tests carried out on internal combustion engines, which examined particle number emissions and the effect of renewable fuel on particle number emissions, gave results similar to the above. As the proportion of renewables increased, particle number emissions decreased. That was only true if the dose was not compensated due to the lower calorific value of ethanol for the same power output [35,36]. If the dose was compensated for bioethanol, the particle number emission increased compared to the original fossil fuel operation [37,41].

*4.3. Operation of the Heater with Diesel*

4.3.1. Comparison of Gasoline and Diesel Heaters

In the product range of the manufacturer of the tested device, the same product can be found in diesel and gasoline versions. Based on the manufacturer's exploded drawings, the machine has no structural differences; the difference is in the article number of the fuel pump, the electronic control unit and the burner basket [65]. Looking further into the manufacturer's parameters, it can be seen, based on Table 3, that there are 12 and 24 V versions of the diesel-powered type, and the manufacturer claims somewhat less consumption with the same heating power.

**Table 3.** Technical data of diesel and gasoline heater [61].

| Technical Data | Air Top Evo 55 | |
|---|---|---|
| | Diesel | Gasoline |
| ECE Approval Number ECE R122 (Heating System) | E1 00 0386 | |
| ECE Approval Number ECE R10 (EMC) | E1 05 5529 | |
| Heat output, control range/boost [kW] | 1.5–5.0/5.5 ** | 1.7–5.0/5.5 ** |
| Fuel consumption, control range/boost [l/h] | 0.18–0.61/0.67 ** | 0.25–0.70/0.80 ** |
| Rated voltage [V] | 12 | 24 |
| Rated power consumption, control range/boost [W] | 15–95/130 ** | |
| Heating air volume flow against 0.5 mbar, control range/boost (m$^3$/h) | 200/220 ** | |
| Fuels *** | Diesel EN 590 B100 FAME EN 14214 [10] HVO DIN EN 15940 [66] | E0-E10 EN 228 |
| Operating temperature range [°C] | −40 to +40 | |
| Dimensions L × W × H [mm] | 423 × 148 × 162 | |
| Weight [kg] | 5.9 | |
| Automatic altitude compensation [m] | 2200 | |

* Increased heat output ("Boost") possible for max. 6 h. ** Increased heat output ("Boost") possible for max. 30 min. *** Other fuels on request.

Since the heat of combustion and calorific value [7,8] of gasoline and diesel per unit mass are almost the same, but that of diesel is greater, the diesel boiler is capable of the same heating performance as the gasoline boiler with less fuel consumption. It can be assumed that the control program differs even in stationary operating conditions. In the case of both versions, the fuel pump results in frequency-controlled, pulsating liquid transport; the use of different pumps can be justified by the chemical and physical properties of the two fuels, such as their density. When examined visually, the burner baskets with glow plugs would be identical; the difference could be in their burning mesh metal fabric. The specification for the fuel of gasoline-powered stationary heaters is E0-E10 EN 228, and the fuel for diesel stoves must comply with the Diesel EN 590, B100 FAME EN 14214, and HVO DIN EN 15940 standards. After examining the above data, the authors were curious as to whether the device, initially designed for gasoline operation, works stably with diesel, and if it does, what the results are regarding particle number emissions. The following subsection presents the results obtained with the tested fuels E10 and B7.

4.3.2. Effect of Diesel on the Particle Number Concentration

During the start-up phase, when the fuel pump starts delivering the fuel and the fuel evaporates through the burning mesh to the working glow plug, for the first 150 s or so of the operating cycle, almost the same particle emissions are observed when comparing gasoline and diesel operation. As the start-up phase was presented in Section 4.2.1, in the case of gasoline, it reaches its stable operating state after approximately 280 s. This value can also be observed in the case of gas oil, but the particle-number-time function develops differently (Figure 15). After a smaller rise and a smaller peak until the 175th second, it decreases by approximately the same amount. In Figure 16, it can also be seen from the air condition factor values that the mixture becomes rich quickly in the case of E10, and the lambda decreases sharply, while a slower enrichment can be observed in the case of B7.

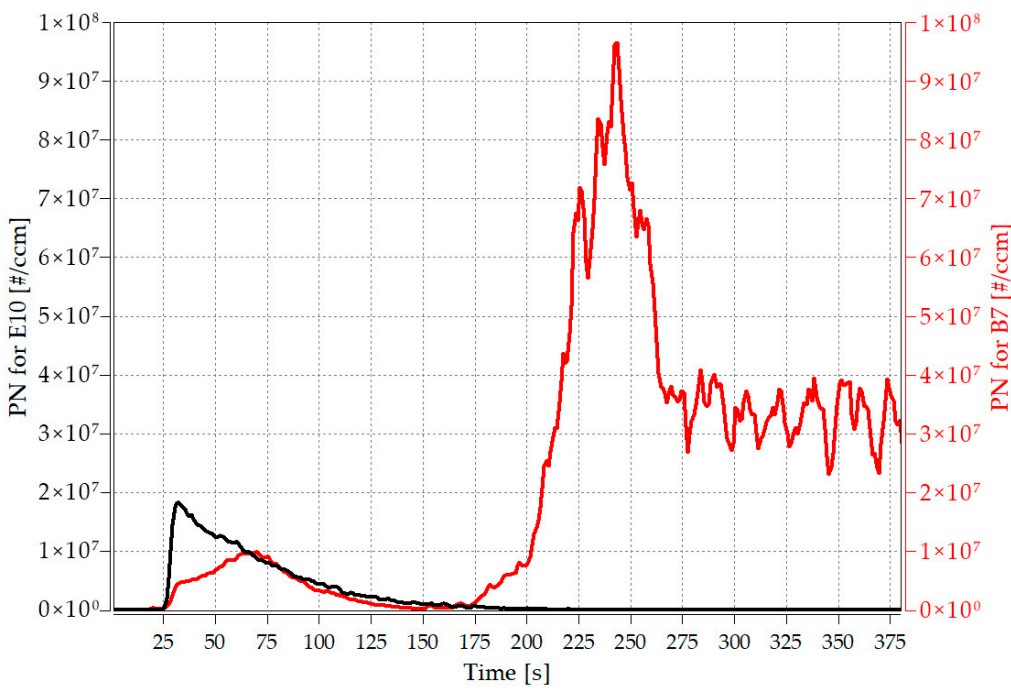

**Figure 15.** Particle number concentration changes for E10 and B7 in the start-up phase.

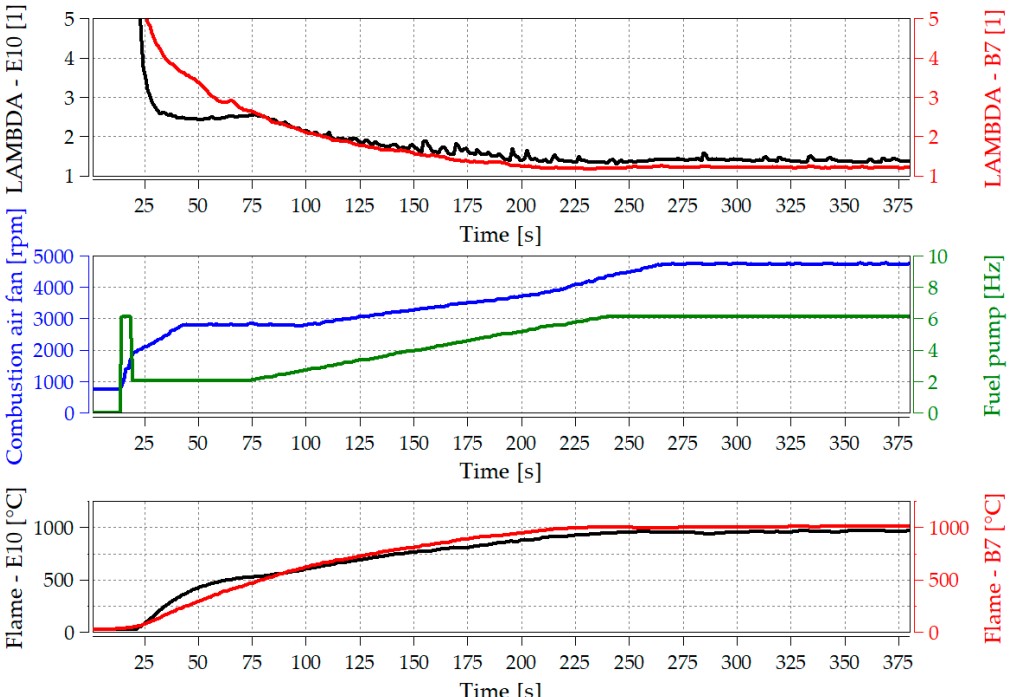

**Figure 16.** Operational parameters of E10 and B7 during the start-up phase (upper part: air factor; middle part: fan and fuel pump; lower part: flame temperature).

After the 175th second, in contrast to operation with gasoline, when the particle emission shows a steady decreasing trend, in the case of diesel, there is an increase between about 170 and 275 s and then a large oscillation, the reasons for which can be, among others, as follows:

(i) Then the lambda has a value of around 1.7 (Figure 16), which is already a critically low value in the case of diesel combustion; the fuel droplets do not receive enough oxygen. Between 250 and 275 s, with the fuel delivery already stable, the air delivery rate increases even further; thus, the lambda rises slightly, with which a slight drop in particle emissions can be measured.

(ii) Diesel has a higher density than gasoline [7,8]; thus, with the same volumetric delivery or pumping frequency, a larger mass of diesel enters the space than gasoline.

(iii) Diesel has a boiling point 100–200 degrees higher than gasoline [67]. That means that at the same temperature, only a part of the diesel (or none) transforms into a vapour state, which is already suitable for ignition.

(iv) As a liquid, diesel oil has a higher surface tension than gasoline [68]. This affects the formation of droplets from the large liquid pool and the breakup of the droplets.

The effect of the four factors listed above may be the lower emissions and the subsequent jump during the 25–175 s interval due to the worse mixture formation with diesel. The explanation is not entirely certain because, in the time range of the large-scale number of emissions, no jump is visible in the flame temperature and the value of the atmospheric factor.

In the stationary state, during the measurement with gasoline, the average value of the lambda was 1.38, in the case of diesel, 1.2, and the particle emission showed an increase of nearly 2000 times under the same operating conditions. A significant jump in the emission of particle numbers is experienced during gasoline combustion when the lambda value increases (Figure 17), as presented earlier. In the steady state, the average value of the particle number emission with B7 is orders of magnitude higher than in the case of E10. That is undoubtedly because the average air factor value 1.2 is too rich for a diesel plant. In the case of the transient process occurring in the steady state, when testing with gas oil, the particle emission is significantly reduced compared to that measured in the stationary state. Then, the value of lambda increases ($\lambda \approx 2$), substantially reducing the particle number emission (Figure 18).

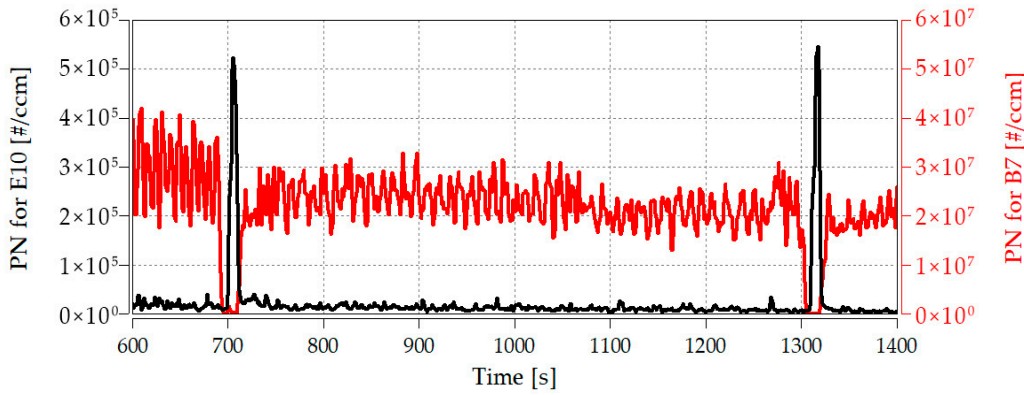

**Figure 17.** Particle number concentration changes for E10 and B7 in the steady-state phase.

The particle number-time functions for the burn-out phase are shown in Figure 19. It can also be seen here that the number of emitted particles is orders of magnitude higher with the burning of B7 in the stable phase. With the end of the fuel supply, the lambda value, which is favourable for the diesel plant and goes in an increasing direction, has a beneficial effect on the particle number-time evolution, and a non-negligible number of particles is still generated from the flame that ends afterwards. The situation is different for diesel. The flame extinction, temperature drop, and change in the air condition factor have a smaller slope (Figure 20). That can be explained by the fact that, in the case of diesel fuel, in contrast to gasoline, parts of the fuel remain in the combustion chamber, and oxidation processes continue to take place with the help of residual heat and airflow (fan).

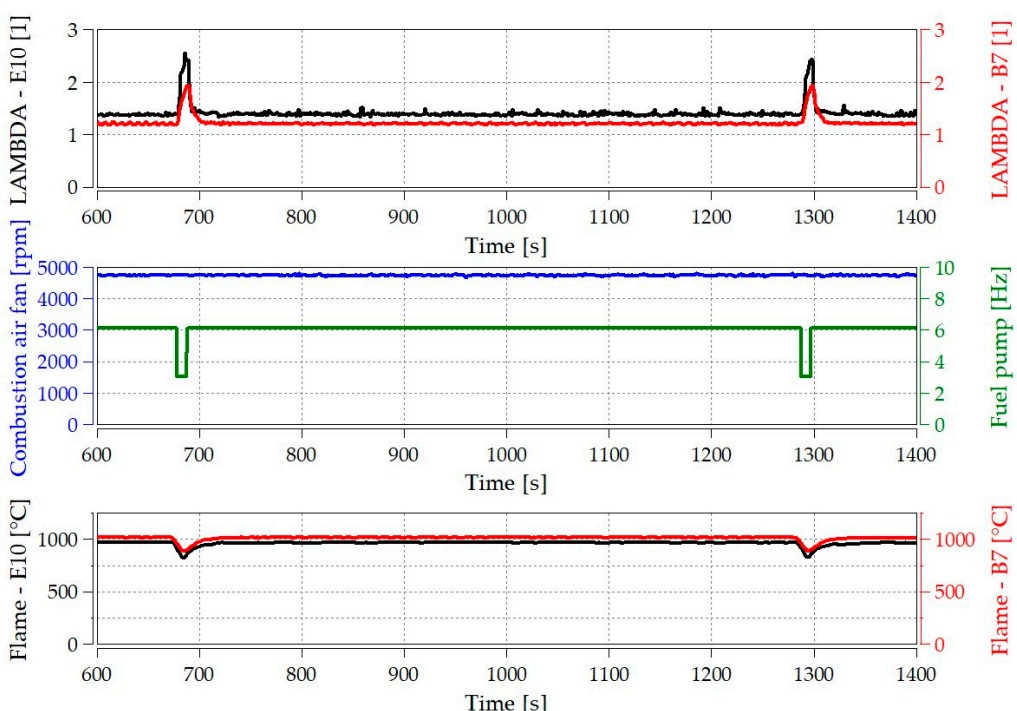

**Figure 18.** Operational parameters of E10 and B7 during the steady-state phase (**upper part**: air factor; middle part: fan and fuel pump; **lower part**: flame temperature).

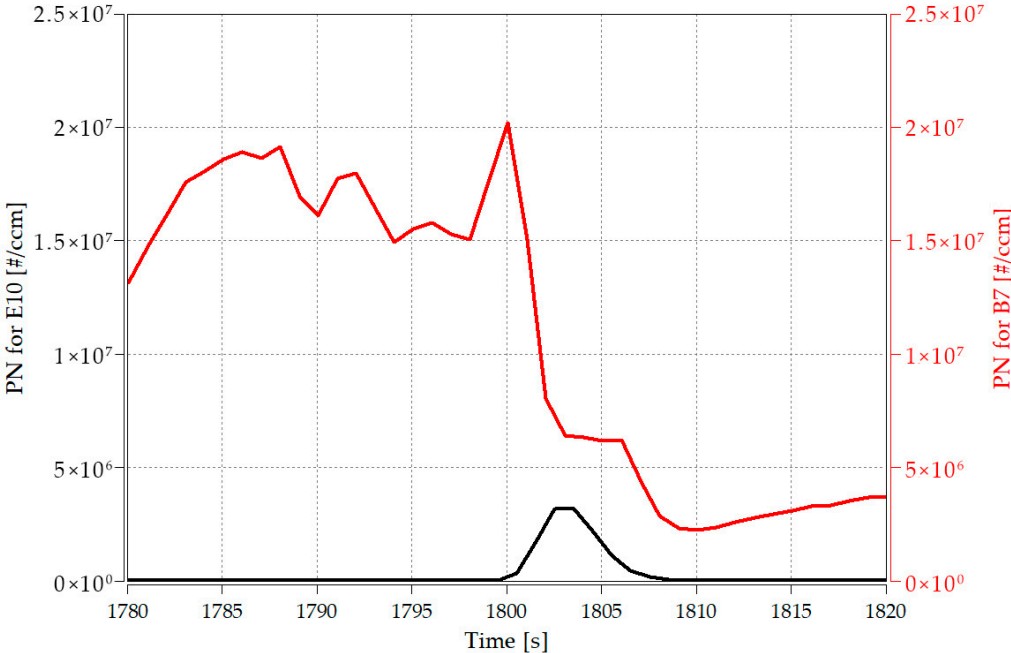

**Figure 19.** Particle number concentration changes for E10 and B7 in the burn-out phase.

The value of the air excess ratio significantly influences the particle number emission. Lower air-to-fuel excess ratio conditions favour particle number emissions [36]. This trend was also visible in the above results. The value of the air condition factor changed during operation with diesel fuel compared to operation with motor gasoline because the doses and air delivery did not change. However, the stoichiometric air condition of diesel differs from that of gasoline [67].

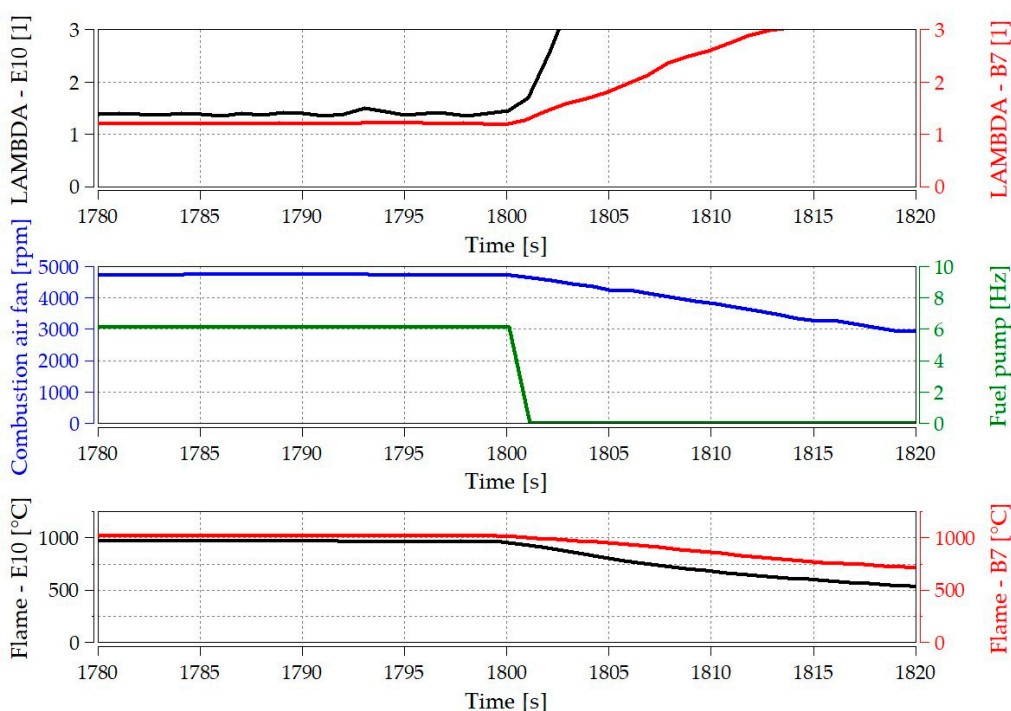

**Figure 20.** Operational parameters of E10 and B7 during the burn-out phase (**upper part**: air factor; middle part: fan and fuel pump; **lower part**: flame temperature).

### 4.4. Depositions in the Chamber and on the Burning Mesh

Figure 21 shows the burner basket in the stationary heating device in a clean state before operation and with deposits after several cycles. These two pictures are just a simple demonstration of the fact that there is a deposit after a long-term process, which can be easily determined by visual inspection.

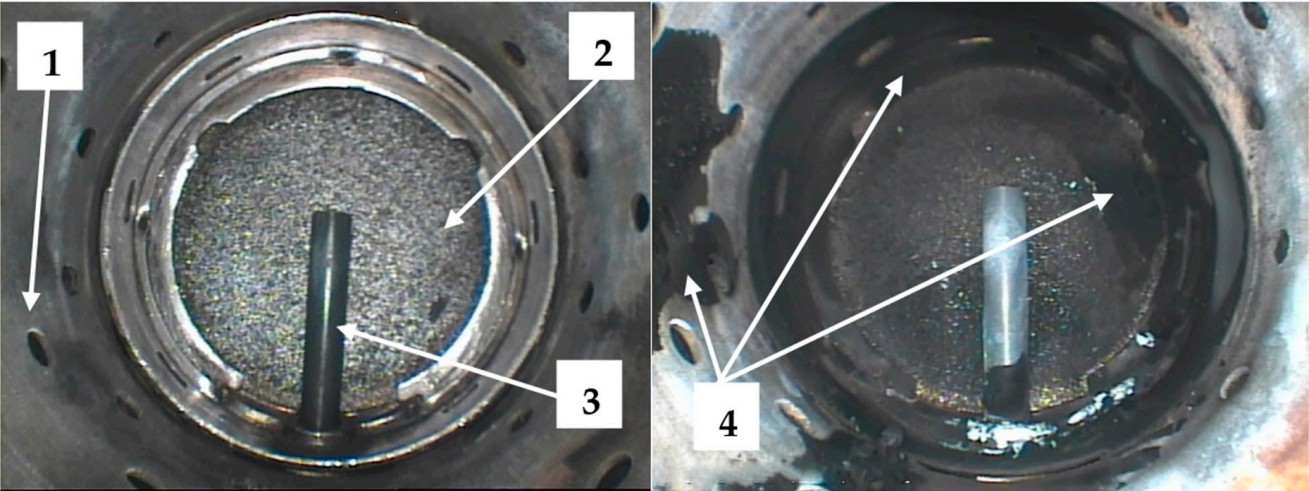

**Figure 21.** Combustion chamber in a clear condition (**left**) and after a period of usage (**right**) (numbering: 1—air holes, 2—evaporator, 3—glow plug, 4—deposits).

With the formation of soot, it is inevitable that deposits also form in the combustion chamber, which creates more significant and larger agglomerates during use. At marking 4, it can be observed that deposits are also formed around the holes for the combustion air in addition to the evaporator. These can narrow the injection cross-section; thus, less air can enter the combustion chamber, thus hurting combustion, which can increase particulate emissions. Deposits on the burning mesh can have both negative and positive effects. It

can have a negative impact if it reduces the cross-section for fuel flow to such an extent that it causes a decrease in heat output. A positive effect may be that if there is a reducing effect on the average hole diameter, the fuel enters the combustion chamber with more pressure and breaks up into smaller droplets as it passes through this mesh. After the tests with different mixtures (3 × 30 min), the stationary heating device was disassembled. After disassembly, we took photographs of the burner baskets shown in Figure 22. for the different tested fuels (E10, E30, E100). The glow stick had already been removed from the device at that time.

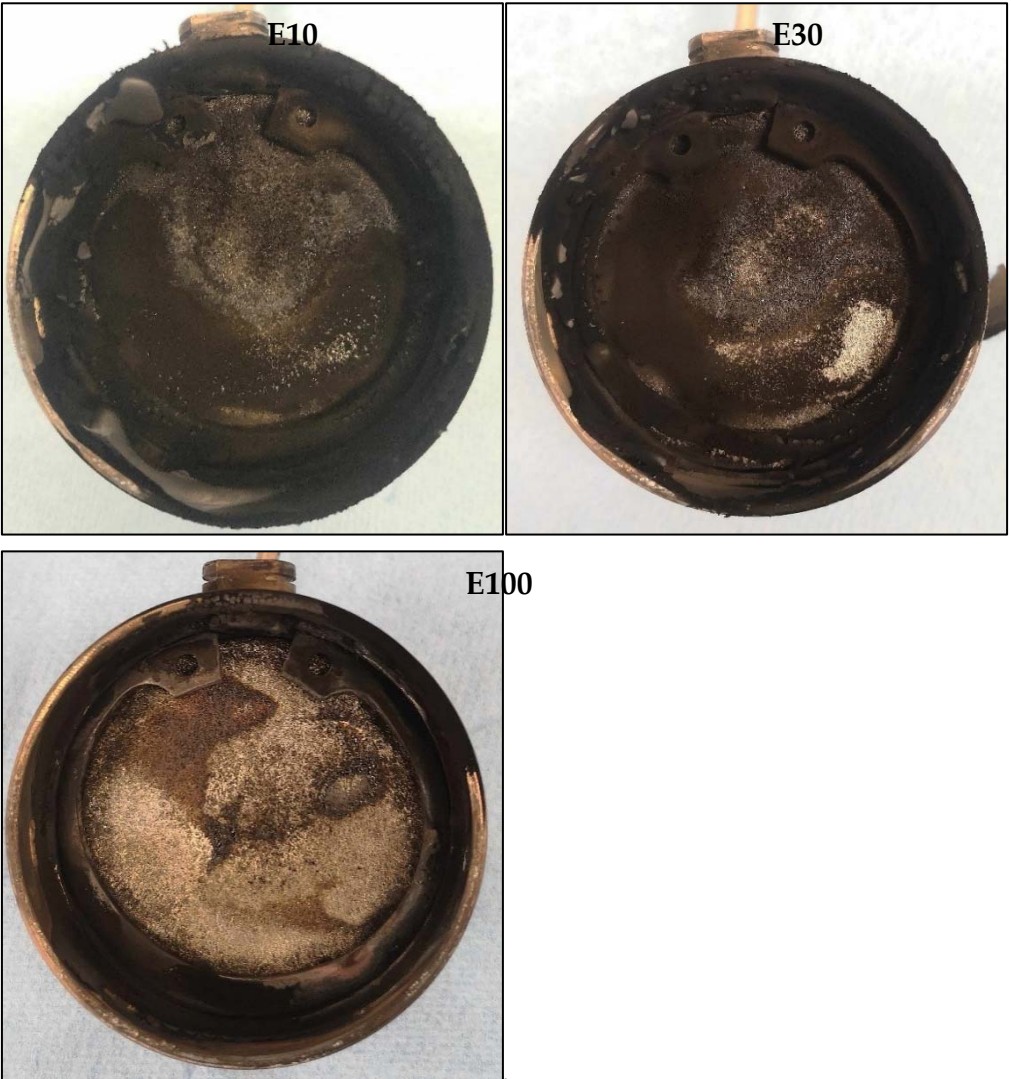

**Figure 22.** The amount of soot deposited in the burner basket as a function of the change in bioethanol—visual inspection.

It is visible that the combustion basket remained cleaner when moving from E10 to E100. That is related to the amount of soot particles produced, in the case of E100, less soot was made, and the combustion was cleaner. The composition of fuels is a significant determinant of the combustion process and, therefore, affects the production of harmful substances, i.e., particle number emissions and deposits. Soot is formed from aromatics of the fuel [69–71]. According to [12], the permissible aromatic content in motor gasoline is less than or equal to 35 V/V%. In the case of diesel, the amount of polycyclic aromatics is limited to 8 m/m% [11]. There are no aromatics in bioethanol, it has a chemically homogeneous structure [9].

*4.5. Analysis of the Chemical Composition of the Deposited Soot*

In the following, the deposits resulting from operation with E10 and E30 fuels were analysed and presented during the elemental analysis. In the case of the other two tested fuels, E100 and B7, we cannot deliver results due to the following:

- During the $3 \times 30$-min cycles during operation with E100, no deposits were formed that were large enough for us to take a sample for elemental analysis.
- After the operation of B7, the sampling was carried out, but due to the failure of the SEM device, we could not perform the analysis. Unfortunately, the device was not repaired before the manuscript was submitted.

In terms of their composition, soot particles are particles consisting of a mixture of unburned hydrocarbons and combustion ash. During the examination of the elemental composition of the energy-dispersive X-ray spectroscopy, the following elements could be identified in each soot sample: carbon, oxygen, fluorine, zinc, calcium, magnesium, chlorine, nitrogen, phosphorus and sulphur. Elements with more than 5 m/m% content were designated as main components in proportion to their occurrence; this included carbon and oxygen. Elements occurring in concentrations between 0.5 m/m% and 5 m/m% were defined as major components, namely fluorine and zinc. Meanwhile, the elements appearing in a proportion of less than 0.5 m/m% were designated as minor components (Ca, Mg, Cl, N, P, S). The measurement results are summarized in Table 4.

**Table 4.** The averages of the normalized mass percentages of the elements found in the three tested soot samples.

|  | C | O | F | Zn | Ca | Mg | Cl | N | P | S |
|---|---|---|---|---|---|---|---|---|---|---|
|  | **[m/m%]** | | | | | | | | | |
| E10 | 92.18 | 6.34 | 0.64 | 0.56 | 0.09 | 0.06 | 0.04 | 0.03 | 0.02 | 0.02 |
| E30 | 88.54 | 9.20 | 1.18 | 0.70 | 0.11 | 0.10 | 0.05 | 0.04 | 0.04 | 0.03 |

The soot generated during the combustion of different ethanol-containing fuels (E10, E30) showed the expected results. Figure 23. shows the distribution of the composition of each soot sample.

Based on examining the elemental distribution of the soot samples, it can be established that its two main components are carbon and oxygen. As the ethanol content of the fuel increased, the carbon content of the soot decreased: between the composition of the E10 and E30 soot samples, a difference of 4.11% by mass can be measured in terms of the soot content. Since the amount of soot that can be collected from the burning mesh decreases as the ethanol content increases. At the same time, a decreasing carbon content can be measured in the soot samples. It can be concluded that when burning fuel with a higher ethanol content, there is less unburnt carbon from the fuel. This can be a consequence of two things. Firstly, ethanol has lower carbon content in its chemical composition than gasoline has, and secondly, ethanol also has oxygen, which is not included in gasoline. Hence, ash occurs in a higher proportion of the particles. On the other hand, the tendency of the oxygen content of the soot increases due to the ethanol content, which also shows a higher proportion of ash in the soot. During the rise of the ethanol content from 10 to 30 V/V%, the oxygen content of the carbon black increased from 6.34% to 9.20% (+2.86%), see Figure 24.

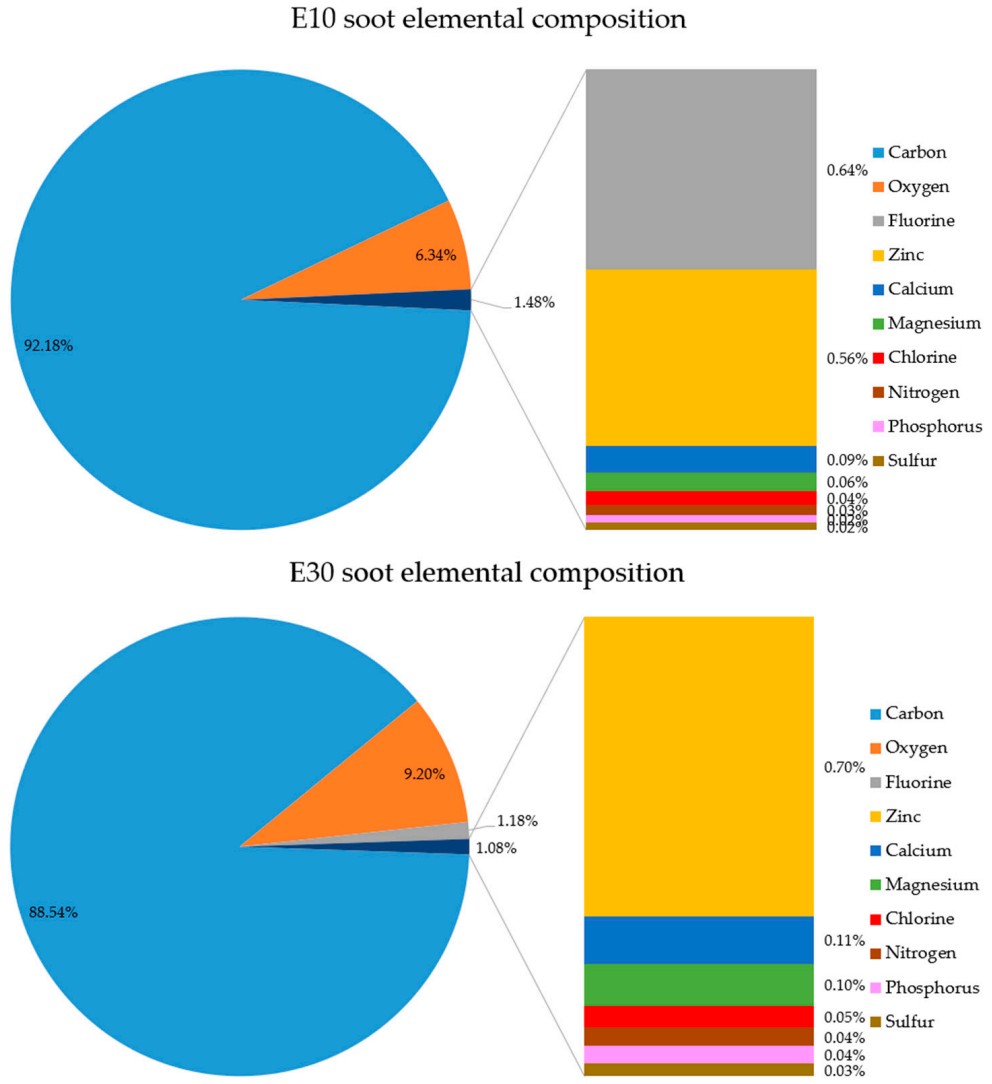

**Figure 23.** The elemental composition of soot particles produced during fuel combustion containing different percentages of ethanol.

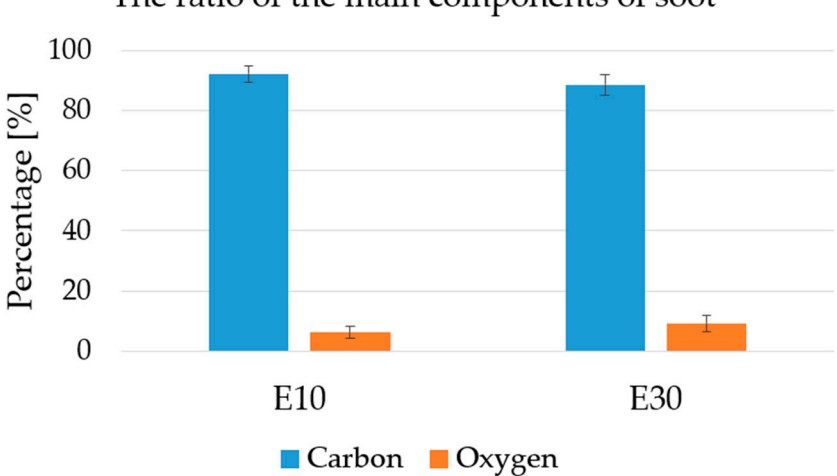

**Figure 24.** Mass percent ratio of carbon and oxygen content of the soot samples compared to the ethanol content of the fuel.

In addition to carbon and oxygen, significant amounts of fluorine and zinc, which may come from the additives of fuel, were found in the soot samples, see Figure 25. The fluorine content of soot increases significantly as the ethanol content increases in the fuel. However, when analysing the results, it is worth considering that the EDS measurement of fluorine is difficult; thus, the standard deviation of the measured results is significant. The zinc content, typically added to the soot during the combustion of lubricating oil additives, shows uniform values in the soot. It is essential to mention that in this case, we did not use lubricating oil, only the tested fuels. Based on the results, no significant correlation can be established between the zinc content of the soot and the ethanol content of the fuel.

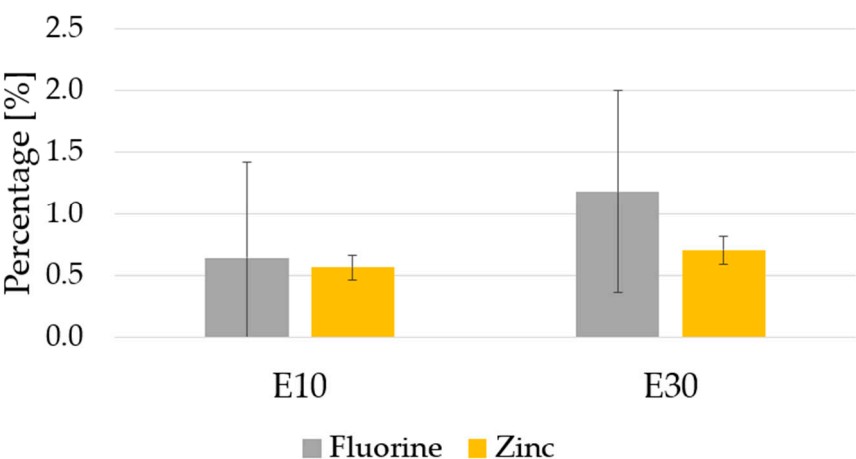

**Figure 25.** The mass percent ratio of fluorine and zinc content of the soot samples was compared to the ethanol content of the fuel.

As the ethanol content of the fuel increases, the minor components in the soot—with the decrease in the carbon content as the main component—all increase, as shown in Figure 26. Calcium is an exception in this trend, the occurrence of which remained constant regardless of the ethanol content. However, the magnesium, chlorine, nitrogen, phosphorus and sulphur content in the soot sample shows a steadily rising trend as the ethanol content of the fuel increases. Similar to the detection of fluorine, the measurement of magnesium and nitrogen values is also problematic; thus, the standard deviation of the values is typically high.

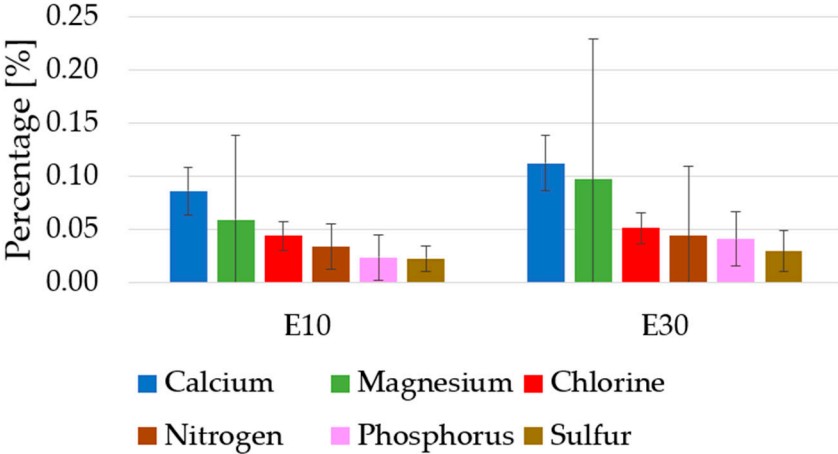

**Figure 26.** Mass percent ratio of the minor components of the soot compared to the ethanol content of the fuel.

Unlike the results of [46], where carbon, oxygen and sulphur were found during the analysis of diesel engine soot, in our case, several other elements (Fluorine, Zinc, Calcium, Magnesium, Chlorine, Phosphorus) were found in the analysed sample. His results agreed with the results of [46,47] in that the sample base is carbon and oxygen. We did not find any test results where the effect of renewable fuel on EDX results was investigated.

### 4.6. Creating and Evaluating Some Particle Number Relevant Parameters

Table 5 shows the number of particle emissions the different tested fuels produced during the 1800 s test periods. The first line shows the mass of fuel consumed during the cycle, which was calculated from the volume delivery of the pump and the weighted average for each fuel and weighted in proportion to the volumes for E30. The density values are as follows: the density of the base fuel was taken as 0.7475 kg/m$^3$ [8], bioethanol as 0.789 kg/m$^3$ according to the datasheet [9], and gas oil as 0.86 kg/m$^3$ [7].

**Table 5.** Particle number relevant parameters.

| Investigated Fuel | E10 | E30 | E100 | B7 | EURO 5-6 Emissions Limits [55] | |
|---|---|---|---|---|---|---|
| Fuel consumption during 1800 s [kg] | 0.2546 | 0.2590 | 0.2690 | 0.2934 | | |
| Particle emission during 1800 s [#/cycle] | $9.56 \times 10^8$ | $4.83 \times 10^8$ | $1.65 \times 10^8$ | $3.92 \times 10^{10}$ | | |
| Particle number per one kilogram of fuel [#/kg × cycle] | $3.76 \times 10^9$ | $1.87 \times 10^9$ | $6.14 \times 10^8$ | $1.34 \times 10^{11}$ | Positive ignition | Compression ignition |
| Particle number per one km [#/km] | $9.56 \times 10^7$ | $4.83 \times 10^7$ | $1.65 \times 10^7$ | $3.92 \times 10^9$ | $6 \times 10^{11}$ | $6 \times 10^{11}$ |

All the values obtained in the second row, measured during one cycle, were determined by testing, the specific values have already been given above. That is not a negligible parameter because it shows that billions of small particles are released into the ambient air from such a small, low-power heater operating under non-forced operating conditions, which heats the air space of vehicles or other cabins in cold weather.

The values in the third row were created for comparing fuels based on a unit mass. According to the results, the values obtained with diesel oil are the highest, followed by E10, E30 and E100. Compared to E10, the E30 mixture shows a −50.35% decrease in particle number emissions, and the E100 −83.66%. Based on the values, diesel is the most polluting fuel based on particle number emissions, and pure bioethanol is the least polluting.

The numerical values shown in line 4 are also created, as well as calculated values. We included this to compare the emission values measured in this research with the limit values of the emission type test of passenger vehicles. The calculation was straightforward, it was obtained by dividing by ten from the values of row 2. The background of the analysis, i.e., the idea of dividing by 10, came from the fact that, according to [27], a vehicle during the emission type test cycle lasted approximately 1800 s and travelled 10 km. According to the values, the particle number emission of the heater is one to two orders of magnitude smaller than the limit value. Considering that this device is "only" a piece of equipment suitable for heating the passenger compartment and used compared to a passenger car, which can also produce heating energy but whose primary purpose is to create traction, such a device is highly polluting. Suppose such a device is installed in a vehicle that does not have an internal combustion engine in its drive system. In that case, that vehicle is at least as polluting as a vehicle powered by an internal combustion engine during passenger space heating use. If such a device is installed in a car with an internal combustion engine, this device creates a massive amount of extra pollution in one heating cycle.

### 4.7. Summary for Results and Discussion

The tests carried out could have been more comprehensive in terms of particle-relevant parameters. To the best of our ability, we selected two critical parameters: the number of

particles and the elemental analysis of the deposit. It would have been possible to perform particle mass, blackening number, particle morphology, or numerical analysis in other particle size ranges, etc.

In their previous article [56] on harmful emissions from heating devices, the authors of the current manuscript examined the emission of gaseous components of such devices. Based on the measurement results, a straightforward model was used to calculate the total emissions of a vehicle fleet (the size of Hungary) equipped with such a device. This shows that for certain components, e.g., $CO_2$, the emission is not significant. However, it concerns other components, e.g., THC (Total Hydrogen Carbon), which is directly harmful to the environment.

Based on the present measurement and calculation results, which refer to the particle number emission, it would be worthwhile to consider the following: First, the particle number emission should also be regulated at the vehicle's type approval level. Our results, measured at 15 °C, are, on average, three orders of magnitude lower than the permissible emissions of a vehicle engine. Research results of [26], which examined the particle number concentration of passenger cars in the cold ($-10$ °C) conditions, are, on average, three orders of magnitude higher than the EURO 5–6 particle number limit values. This device must be a heater with a power output of a few kW heat performance, compared to a vehicle with an engine of more times ten or more than 100 kW. Thus, these devices are highly polluting about their unit performance. The second area is the calculation of total emissions from transportation. Certainly, the emissions of these devices are not included in the total transportation emissions, even though it would be reasonable in terms of the particle number parameter just obtained. It can probably be achieved with the help of a properly designed vehicle fleet model. If there is a received value, it can be used to decide whether to intervene. This effect is still "with us" and pollutes the environment if we do not calculate its extent. The third point concerns electric vehicles. If an electric vehicle is equipped with such a device and in use, that electric vehicle is by no means emission-free.

## 5. Conclusions

In the previously presented research, we presented the results of a detailed particle number emission and soot element analysis of a stationary heating device used in vehicles operated with a base fuel (E10), another gasoline–ethanol mixture (E30) and pure ethanol (E100) and diesel (B7). Based on our results, the most important conclusions are as follows:

- During the measurement of the number of emitted particles, the three operating phases of the device can be distinguished: the start-up, steady-state and burn-out phases. Overall, 95% of the total particle number emissions that can be measured during the entire 1800 s measurement cycle occur in the start-up phase. It can be concluded that it is necessary to strive to use the heater in a stable operating state for as long as possible and to avoid intermittent operation.
- By increasing the V/V% bioethanol content in the tested fuel, the air factor also shows a higher value. As a result, the oxidation of fuel droplets is promoted, so the particle emission is significantly reduced. Compared to the E10 fuel used today, 95% fewer particles can be measured using E100 during stable operation.
- In the case of the three tested fuels, the total particle emissions of the start-up phases correspond to approximately 90 h for E10, 70 h for E30, and 295 h for E100 in stable working conditions.
- The stationary heating device designed for operation with gasoline also works stably with diesel. Since our device was optimized for burning gasoline, the lambda value was reduced by introducing a similar amount of fuel and air when burning diesel, the mixture was extremely rich in fuel, as a result of which an order of magnitude higher particle emission could be measured.
- During the examination of soot samples, it can be established that the two main components of soot are carbon and oxygen atoms. As the ethanol content increases from 10 to 30%, the carbon content of soot decreases and its oxygen content increases;

thus, we can conclude that the combustion process is more effective when burning fuel with a higher ethanol content. Fewer soot deposits are visible in the burning mesh, and the amount of non-organic content (ash) has increased. It is essential to note that in the case of our measurements, non-organic components can only come from the fuel, compared to an internal combustion engine, where they can also come from engine oil and metal wear.

- Based on the specific indicators created for the measured particle numbers, operating with diesel fuel is the most polluting and pure bioethanol is the least polluting per unit mass of fuel. As for the absolute results of particle number emissions, depending on the fuel, they are one or two orders of magnitude lower than the limit values included in the type test regulations for passenger vehicles. It is a highly polluting device because it is "only" a heater compared to a car engine, generating a significant driving force.

## 6. Outlook

At the end of the Results and Discussion section, we covered a lot of things, including how this topic can be broadened and deepened. At our practical level, where we can perform measurements in a laboratory, an exciting direction for us is the relationship between the heating power available with renewable fuels and the heating power available with the original fuel. Furthermore, how many additional emissions are caused by achieving the same heating performance with the renewable fuel or a mixture containing a higher amount?

**Author Contributions:** Conceptualization, P.N. and G.S.; methodology, P.N. and G.S.; formal analysis, P.N.; investigation, P.N. and Á.I.S.; resources, P.N.; data curation, P.N. and Á.I.S.; writing—original draft preparation, P.N.; writing—review and editing, G.S.; visualization, P.N. and Á.I.S.; supervision, I.Z. All authors have read and agreed to the published version of the manuscript.

**Funding:** This research received no external funding.

**Data Availability Statement:** Data available on request. The data presented in this study are available on request from the corresponding author.

**Conflicts of Interest:** The authors declare no conflicts of interest.

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
