# Peer review of "Particle Number Concentration and SEM-EDX Analyses of an Auxiliary Heating Device in Operation with Different Fossil and Renewable Fuel"

_inventions, doi:10.3390/inventions9010013_

Round 1

Reviewer 1 Report

Comments and Suggestions for Authors

This is an interesting manuscript that falls within the scope of the journal. Please find below comments to further improve this work:

-Fig. 3 – make sure you have the copyrights for this, I think you could also put this in the introduction already (up to you, showing it later as the specific heater design you examine is also fine I guess)

-Can the pre-written programs influence the emission levels? Are they tailored for the use of specific fuels? I think this needs to be elaborated better.

-Fig. 4 caption check language

-I feel that there is a lot of valuable information in this work. I also feel that it is a bit lengthy and you can probably present the information in a more concise manner.

Author Response

Dear Reviewer,
The authors are grateful for your comments. They are helpful and constructive for us. We took  each comment and suggestion in order and made changes in the manuscript accordingly, and then we responded to each of them. Responses to each comment can be found in a separate file, attached. Changes made in the manuscript have been highlighted with blue. 

Reviewer 2 Report

Comments and Suggestions for Authors

1) In the Abstract the Authors should add more the most important results obtained in this research (their exact values), a rough description only is not sufficient. Such addition will highlight the paper novelty already in the Abstract.

2) The Authors should add in a paper at least a general specifications of each tested fuel.

3) In the paper should be added an exact picture of an experimental setup. According to the presented explanations, it seems that the Authors have performed experiments on the existing experimental setup, but the picture (with marked the most important elements and measuring devices) will remove any concern. I must highlight this because in many papers which I have reviewed so far, the Authors have explained experimental setup and measurement devices which did not really exist or which are taken from the literature – therefore, the experimental setup picture will resolve any concern. This element is very important because the whole paper is based on the performed experiments.

4) Section 3 title should be corrected – “Experimental” only is not a proper Section title.

5) Through the paper text can be found obvious and typing mistakes – all of them should be corrected during the revision process. For example – Figure 4 title – what “és” means? This is just one example, similar obvious and typing mistakes should be corrected throughout the paper.

6) Please, be consistent in the paper. For example, calls on the Figures should be unified – sometimes is Figure 19, sometimes Fig. 19. Also, call on the figure should be exact in the paper text because something is used call Figure 11, sometimes Diagram 11. The modifications are required throughout the paper text.

7) Conclusions – what does it mean the last sentence in the Conclusions – “This section is not mandatory but can be added to the manuscript if the discussion is unusually long or complex.”? The modifications are required.

8) Section 6. Outlook – in this section the Authors should briefly describe the guidelines in further research. At the moment, this Section is too general and overall, it did not offer proper guidelines in further research related to this research topic.

9) The English is understandable, but it can be improved in many sentences. Please, perform a careful check of the English throughout the paper text.

Final remarks: This is a very interesting paper and well-performed research. All my above mentioned comments are actually minor ones, but I believe that they should be involved in the paper during the revision process with an aim to improve paper before publication on the best possible way.

Comments on the Quality of English Language

The English is understandable, but it can be improved in many sentences. Please, perform a careful check of the English throughout the paper text.

Author Response

(The authors gave the same response as above.)

Reviewer 3 Report

Comments and Suggestions for Authors

The manuscript deals with an scientifically challenging topic but there is still room for fundamental organizational, argumentation and structural improvements prior it to be accepted for publication. To this end the following review comments can be considered.

1. It is morphological correct and complete if authors break the lengthy narrative block of section 1, having also developed the missing section 2. Literature Review. This section can be focused on the: “technological”, “environmental”, “normative, regulatory, legislative, directories, measures, recommendation” evolution that has been globally reported in the fields of:

-“an auxiliary heater. They can be classified according to whether they work with diesel or gasoline and whether they heat water or air”.

-the role of bioethanol and generally the renewable fuels.

For them, up to one and cross-citing text page of this new section 2. Literature Review is adequate. Besides, the subdivision of the whole section 1 into 2-3 shorter subsections within it, is advisable.

2. The lumping of citations is an unacceptable practice of enriching the theoretical overview with non-discussed studies. It is particularly noted that the following vaguely-perceived and abstractly-approached terms to be considerably expanded by at least half a page each, having also grouped citations no more than groups of 2-3 citations per paragraph:

….…….and Otto engines [35-44].

….…..decreases when renewable fuel is mixed in [35-44].

….…magnitude higher than in the exhaust gas of Otto engines [29-44].  

….…there are moving parts in the machine or not [24-44].

All these text extracts are statements of repetitions of the same information, but it is impossible all these 10-20 citations to coincide mentioned the “same thing”. There should be somehow differentiation points among them. Please -very please- refer to them only at once, no multiple times, conveying what piece of distinct and separate information you want to point out at each one of these dispersed text points through which these four text extracts are referring  each time.

3. Almost all section 4 and its subsections, lines 269-681, contain analysis and results which are not cross-cited (they are deprived from citations), thus, weaken the validity and the verification of the conducted analysis and findings. If possible a more rigorous and systematic cross-citing of them is needed, including Figures’ more intensified interpretation, and Tables-data’ meaningfulness.

4. Section 6 has to be transferred and incorporated in one narrative under the “Discussion” section. The critical point in the “Discussion” part of the analysis is authors to be based on, but to also forecast and extrapolate the significance of their findings towards a wider environmental-technological-driven conclusions of generalized truth and (global) applicability to be drawn (beyond the tested materials/conducted experimental sessions).

Author Response

(The authors gave the same response as above.)

Round 2

Reviewer 3 Report

Comments and Suggestions for Authors

The manuscript has been thoroughly and constructively revised, having all research parts fully developed and coverage of the research outcomes in an insightful manner. In this respect, the revised manuscript sustains novel characteristics of analytical, experimental, and widely environmental-energy interest, thus, it can be accepted for publication at the Inventions journal as is.